# Learning to cooperate with emergent reputation via multi-agent reinforcement learning

## Abstract

Reputation, the aggregation of peer assessments diffused through social networks, is a pivotal mechanism for promoting cooperation in social dilemmas ubiquitous to distributed multi-agent systems comprising agents with limited perception and cognitive capabilities. Exploring efficient reputation systems, comprising reputation assessment rules and reputation-based policies, is a long-standing challenge. Previous work assumes predefined reputation assessment rules or models reputation as an intrinsic reward to learn policies, compromising the methods' ability for generalization and adaptation. To address this, we propose a distributed multi-agent reinforcement learning method **COOPER** (**COOP**eration with **E**mergent **R**eputation), which jointly learns reputation assessment rules and reputation-based policies entirely from environment rewards. Notably, leveraging the underlying mechanisms of reputation, we deliberately design the constituent modules of **COOPER** and the data flows among them, overcoming the latency and noise in the feedback signal, caused by the deep entanglement between reputation and policy. Experiments on the donation game and the coin game in grid world environments demonstrate that **COOPER** effectively adapts to various existing reputation systems and co-players. Furthermore, we observe the co-emergence of reputation norms and cooperation in self-play settings. These results hold robustly across diverse social network topologies, underscoring the generalizability and efficacy of our approach.

## 1 Introduction

Distributed multi-agent systems (MAS) have attracted considerable attention for their advantages in scalability, robustness, and efficiency when addressing complex real-world problems (Zhang et al., 2021; Ning & Xie, 2024; Maldonado et al., 2024). These systems leverage decentralized decision-making to harness the collective intelligence of multiple autonomous agents, enabling effective solutions across various domains (Oliehoek & Amato, 2016; Jin et al., 2025; Hady et al., 2025). However, agents' autonomy often implies their pursuit of self-interest. Coupled with limited perception (partial observation) and bounded cognitive capabilities, this can give rise to social dilemmas, where the individually optimal strategies conflict with the collective optimum (Axelrod & Hamilton, 1981; Hardin, 1998; Vlassis, 2007). For example, in unmanned aerial vehicle (UAV) formation tasks, multiple UAVs minimizing their own energy use may target the same position, thereby compromising the group's objective of covering all locations efficiently (Yun et al., 2022). Therefore, developing distributed multi-agent reinforcement learning (MARL) methods that effectively mitigate social dilemmas is of critical importance.

To tackle this problem, reputation mechanisms have emerged as a promising solution inspired by human societies (Nowak & Sigmund, 2005). In human interactions, an individual's reputation is an aggregation of others' assessments regarding that individual's behavior, and it is diffused on social networks via gossip (Nowak & Sigmund, 1998). Humans routinely tailor their interaction strategies based on the reputations of others, and the awareness of being judged incentivizes individuals to act in ways that preserve their own reputations for long-term benefit (Fehr & Fischbacher, 2004). Consequently, reputation systems help promote cooperation in social dilemmas by encouraging agents to forgo short-term gains in favor of sustained collective outcomes. However, existing MARL approaches incorporating reputation often rely on predefined reputation assessment rules (Anastassacos et al., 2021; Smit & Santos, 2024) or model reputation as an intrinsic reward signal (Ren et al.,

2025). By doing so, these approaches simplify the learning problem by assuming a pre-existing reputation norm, such as the desirability of a higher reputation or specific rules for reputation assignment. However, this simplification comes at a cost: such methods may lack adaptability in novel environments or when interacting with unfamiliar agents, and they fail to learn a reputation norm from scratch in a fully decentralized manner (Cordova et al., 2024). Actually, learning such a norm is challenging, particularly in self-play settings where no ground-truth standard exists to guide the emergence of reputation assignment rules that foster cooperation.

To conquer this challenge, we propose COOPER (COOPeration with Emergent Reputation), a novel MARL algorithm that jointly learns a reputation assignment rule and a reputation-based policy in a fully decentralized manner. COOPER comprises two key modules: (1) a reputation assignment module that dynamically integrates assessments from neighbors with direct interaction experiences to assess others and infer how one is perceived by the group; and (2) a reputation-based policy module that leverages reputation assessment for co-players and the estimation for self-reputation to guide actions. Our approach distinguishes itself through its sophisticated module and information flow design, which enables the simultaneous emergence of reputation norms and cooperative policies purely through environmental feedback, without relying on predefined reputation semantics or intrinsic rewards. This endows COOPER with strong adaptability to diverse reputation norms and co-players. Through extensive experiments in diverse matrix games and grid-world environments, we demonstrate COOPER's effectiveness in achieving sustained cooperation across various network structures, its robustness in self-play scenarios, and its adaptation capabilities when interacting with agents with existing reputation norms. Furthermore, we provide a detailed analysis of the emergent reputation norms, offering insights that bridge MARL behaviors with theoretical models of reputation and cooperation. In summary, our key contributions are:

- We introduce COOPER, a novel MARL algorithm that jointly learns a reputation assignment module and a reputation-based policy, enabling the emergence of reputation norm and cooperation without predefined reputation semantics or update rules.

- We empirically demonstrate that COOPER sustains cooperation across diverse network structures and successfully adapts to various co-players and pre-existing reputation norms.

- We provide a detailed analysis of the emerged reputation norms, bridging the gap between MARL behaviors and theoretical reputation models.

## 2 RELATED WORK

The fact that individuals help others based on their reputation is a powerful explanation for large-scale human cooperation (Nowak & Sigmund, 2005; Milinski, 2016). Seminal work like image scoring (Nowak & Sigmund, 1998), showed how simple reputation systems can promote cooperation, inspiring extensive research into reputation norms. A key finding is that higher-order norms, such as *standing* (Panchanathan & Boyd, 2003) and *judging* (Ohtsuki & Iwasa, 2004), which condition assessments on the co-player's reputation, lead to more robust and widespread cooperation. This research culminated in the "leading eight" norms, a family of evolutionarily stable assessment rules (Ohtsuki & Iwasa, 2006). Beyond assessment rules, reputation propagation, often modeled as gossip (Wu et al., 2016; Ellwardt, 2019), and social network structures (Watts & Strogatz, 1998; Barabási & Bonabeau, 2003) further shape the efficiency and stability of cooperation. However, these models fail to establish a reputation norm and reputation-based cooperation from scratch.

Fostering cooperation in self-interested RL agents is a long-standing challenge, particularly in mixed-motive social dilemmas like the Iterated Prisoner's Dilemma or Donation Game, where individual and collective interests are misaligned (Fatima et al., 2024; Jiang et al., 2024). While early work in two-agent settings identified strategies like tit-for-tat (Press & Dyson, 2012), scaling these to large populations is hindered by non-stationarity from simultaneous learning (Du et al., 2023). Recent methods often use centralized training (Leibo et al., 2021) or agent modeling (Rabinowitz et al., 2018), but their reliance on central coordination or reward engineering limits decentralization and adaptability.

Integrating reputation mechanisms into MARL is a promising avenue for addressing social dilemmas. A common approach is to **implement predefined reputation assignment rules** (e.g., image scoring or a norm from evolutionary theory) and provide reputation assessment as input to the agent's policy. Studies by Anastassacos et al. (2021) and Ren & Zeng (2023) have demonstrated that agents

can learn effective strategies conditioned on such pre-defined reputation signals. Similarly, Smit & Santos (2024) showed that predefined reputation rules can foster not only cooperation but also fairness. Another line of work uses **reputation as an intrinsic reward** to guide agents toward cooperative behavior (Ren et al., 2025). This approach can be sensitive to the chosen reward function, requiring careful balancing between extrinsic and intrinsic rewards to avoid unintended behaviors.

## 3 Problem Formulation

We study a networked multi-agent system denoted by a static undirected graph $G = (N, E)$. Here, $N = \{1, \ldots, n\}$ is the agent set. Every agent $i \in N$ maintains a private assessment of all agents' reputations, represented by $\boldsymbol{\xi}_i = (\xi_{i \to 1}, \ldots, \xi_{i \to n})$, where $j \in N$ and $\xi_{i \to j} \in [-1, 1]$, and this assessment is shared with $i$'s neighbors in the network, denoted as $\mathcal{N}_i = \{j \mid \{i, j\} \in E\}$. Each agent's reputation is defined and continuously updated by the collective assessments of its peers. Besides disseminating information, agents are randomly paired to play games and receive rewards. We denote agent $i$'s co-player at time $t$ as $g^t(i)$.

An agent's assessment of others is constantly updated based on (1) reputational information shared by neighbors and (2) her observation in physical interaction with co-players. Meanwhile, agents also adjust their own behavior to maintain a favorable reputation for future benefits. We formalize this reputation-based sequential decision-making problem as a partially observable Markov game:

$$\mathcal{MG} = (N, \mathcal{S}, \{\mathcal{A}_i\}_{i \in N}, \mathcal{T}, \{\mathcal{O}_i\}_{i \in N}, \{\Omega_i\}_{i \in N}, \{\mathcal{R}_i\}_{i \in N}, \{\boldsymbol{\xi}_i\}_{i \in N}, \{\boldsymbol{\mathcal{H}}_i\}_{i \in N}, \gamma)$$

More specifically, the state $s^t \in \mathcal{S}$ is defined as $s^t = (s_p^t, G)$ where $s_p^t$ denotes the physical state, like the grid world observation, and $G$ represents agents' social network status. At each timestep, agent $i$ receives observation $o_i^t \in \mathcal{O}_i$ generated by the observation function $\Omega_i(o_i^t | s^t)$. Agent $i$ takes action $a_i^t \in \mathcal{A}_i$ based on observation $o_i^t$, assessment for others $\boldsymbol{\xi}_i^t$, and her neighbors' assessments $\boldsymbol{\xi}_{\mathcal{N}_i}^t = \{\boldsymbol{\xi}_j^t | j \in \mathcal{N}_i\}$. After joint action $\boldsymbol{a}^t = (a_1^t, \ldots, a_n^t) \in \mathcal{A}_1 \times \cdots \times \mathcal{A}_n$, state $s^t$ transits to $s^{t+1}$ with probability $\mathcal{T}(s^{t+1} | s^t, \boldsymbol{a}^t)$, and agent $i$ receives reward $r_i^t = \mathcal{R}_i(s^t, \boldsymbol{a}^t)$. $\mathcal{H}_i^t = (\mathcal{H}_{i,1}^t, \ldots, \mathcal{H}_{i,n}^t) \in \boldsymbol{\mathcal{H}_i}$ represents agent $i$'s interaction histories with the co-players.

Through interaction, each agent $i$ learns a reputation-based policy $\pi_i(a_i^t | o_i^t, \boldsymbol{\xi}_i^t)$ and a reputation assignment function $u_i(\boldsymbol{\xi}_i^{t+1} | \boldsymbol{\xi}_i^t, \boldsymbol{\xi}_{\mathcal{N}_i}^t, \mathcal{H}_i^t)$ together. The policy $\pi_i$ is optimized to maximize a discounted return $G_{\pi_i} = \mathbb{E}_{\boldsymbol{a}^t \sim \boldsymbol{\pi}, s^{t+1} \sim \mathcal{T}(s^t, \boldsymbol{a}^t)}[\sum_{t=0}^{\infty} \gamma^t \mathcal{R}_i(s^t, \boldsymbol{a}^t)]$ where $\gamma \in [0, 1]$ is the discounted factor. In addition, $u_i$ is optimized to generate a more accurate evaluation for different co-players, which ultimately contributes to a higher return. We say that a reputation norm emerges in a multi-agent system if the individually learned reputation assignment functions $\{u_i | i \in N\}$ converge toward a consistent evaluation strategy, reflecting a shared assessment pattern across the population.

## 4 Methodology

To promote cooperation and adapt to unknown scenarios in mixed-motive games, we propose a distributed MARL method named **COOPER** (***COOP**eration with **E**mergent **R**eputation*), which *jointly* learns (i) a reputation norm implemented as differentiable reputation assignment rules and (ii) a reputation-based policy, *entirely* from extrinsic rewards. Unlike previous work that relies on predefined reputation assignment rules or uses reputation as an intrinsic reward, COOPER derives its learning signal from interaction rewards. This enables the co-emergence of the reputation norm and cooperation, making COOPER highly adaptable to diverse environments and co-players.

As shown in Figure 1, COOPER consists of a reputation assignment module and a reputation-based policy. The reputation assignment module comprises two key components: the *gossip-based* reputation assessment $\psi$ that aggregates neighbors' opinions and the *interaction-based* reputation assessment $\phi$ that refines beliefs using direct interaction histories. This dual-component design captures the feature of human social reasoning and effectively balances social opinions with personal experiences, thereby enhancing the robustness and reliability of the reputation assessment. The reputation-based policy $\pi$ conditions on these assessments to implement farsighted behavior in mixed-motive games, where myopic strategies can exploit short-term gains at the expense of future cooperation. To achieve sustainable long-term cooperation, $\pi$ leverages reputation assessments of co-players to adapt to heterogeneous opponents and leverages the estimation of one's own reputation to regulate its own behavior, guiding agents to account for the future consequences of current actions.

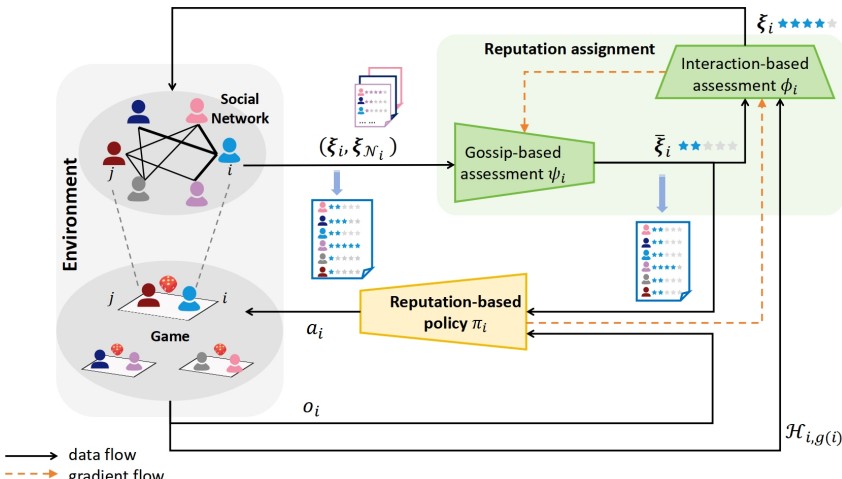

Figure 1: An overview of our method. **COOPER** agents promote cooperative behavior in multi-agent reinforcement learning by jointly learn a reputation-based policy $\pi$ and a reputation assignment module which separately processes gossip-based reputation assignment ($\psi$) and interaction-based reputation assignment ($\phi$). During rollouts, the execution order is $\psi \rightarrow \pi \rightarrow \phi$, and in optimization, the order is $\psi \rightarrow \phi \rightarrow \pi$ to facilitate the learning of an aligned reputation assignment rule and reputation-based policy.

More specifically, at time $t$, each agent $i$ aggregates social information to generate post-gossip assessments $\bar{\boldsymbol{\xi}}_i^t = \psi_{\theta_i}(\boldsymbol{\xi}_i^t, \boldsymbol{\xi}_{\mathcal{N}_i}^t)$. Given $i$'s current co-player $g^t(i)$, actions are then drawn from the reputation-based policy conditioned on the current observation and the post-gossip reputations: $a_i^t \sim \pi_{\theta_i}(\cdot \,|\, o_i^t, \bar{\boldsymbol{\xi}}_i^t)$. After acting, the joint action $(a_i^t, a_{g^t(i)}^t)$ is appended to the history $\mathcal{H}_{i,g^t(i)}^t$, which, together with $\bar{\boldsymbol{\xi}}_i^t$, feeds into $\phi$. The interaction-based module $\phi$ updates agent $i$'s assessment for its co-player $g^t(i)$ as $\xi_{i \rightarrow g^t(i)}^{t+1} = \phi_{\theta_i}\left(\bar{\xi}_{i \rightarrow g^t(i)}^t, \mathcal{H}_{i,g^t(i)}^t\right)$ where $\xi_{i \rightarrow j}^{t+1} = \bar{\xi}_{i \rightarrow j}^t$ for all $j \neq g^t(i)$.

For brevity, when unambiguous we write $\boldsymbol{\xi}_i^{t+1} = \phi_{\theta_i}(\bar{\boldsymbol{\xi}}_i^t, \mathcal{H}_{i,g^t(i)}^t)$, though only the $g^t(i)$-entry is updated. To summarize, agent $i$ updates assessments for others by $u_i = \phi_{\theta_i}(\psi_{\theta_i}(\boldsymbol{\xi}_i^t, \boldsymbol{\xi}_{\mathcal{N}_i}^t), \mathcal{H}_i^t)$. The updated assessments then diffuse through the social network $G$.

The co-learning of the reputation assignment module and the reputation-based policy is challenging because these two are deeply interdependent. The policy depends on accurate reputation assessments (reflecting certain reputation norms) to make farsighted decisions, while the reputation assignment module requires policy outcomes (such as cooperation or defection) to assign accurate reputations. Without careful coordination, this can lead to unstable learning dynamics or failure to converge.

COOPER addresses this challenge through carefully designed modules that capture the fundamental principle of reputation, as well as an alternating optimization scheme that preserves end-to-end training guided by environment rewards. During rollouts, the policy $\pi$ conditions on the pre-interaction assessments of the current co-player. During optimization, we reverse the flow: $\pi$ is trained on the post-interaction assessments computed by $\phi$, based on the latest (just-observed) interaction history. Hence, gradients from rewards and regularizers propagate through $\pi$ into $\phi$ and $\psi$. By aligning updates with the most recent interactions, COOPER grounds training in the most relevant information, enabling more accurate decision-making. It also ensures that $\psi$ and $\phi$ are trained to produce assessments that more accurately predict the co-player's behavior and improve future action selection. Concretely, we formulate the loss function to train $\pi_{\theta_i} \rightarrow \phi_{\theta_i} \rightarrow \psi_{\theta_i}$ as

$$\mathcal{L}(\theta_i) = \mathcal{L}_{\text{env}}(\theta_i) + \lambda_{\text{conf}} \mathcal{L}_{\text{conf}}(\theta_i) + \lambda_{\text{ent}} \mathcal{L}_{\text{ent}}(\theta_i). \tag{1}$$

The first term, $\mathcal{L}_{\text{env}}(\theta_i)$ guides COOPER to jointly learn the reputation assignment module $\psi, \phi$ and the reputation-based policy $\pi$ to maximize environmental rewards. It is formulated as

$$\mathcal{L}_{\text{env}}(\theta_i) = \mathbb{E}\left[\sum_{t=0}^{T}\left(-\hat{A}_i^t \log \pi_{\theta_i}\left(a_i^t \,|\, o_i^t, \, \phi_{\theta_i}(\psi_{\theta_i}(\boldsymbol{\xi}_i^t, \boldsymbol{\xi}_{\mathcal{N}_i}^t), \mathcal{H}_{i,g^t(i)}^t)\right)\right)\right]. \tag{2}$$

We write the policy input as $\phi_{\theta_i}(\psi_{\theta_i}(\boldsymbol{\xi}_i^t, \boldsymbol{\xi}_{\mathcal{N}_i}^t), \mathcal{H}_{i,g^t(i)}^t)$ to emphasize that the interaction-informed, post-gossip assessments drive action selection. In Equation 2, $\hat{A}_i^t = \sum_{t'=t}^{T-1}(\gamma\lambda)^{t'-t}(r_i^{t'} + \gamma V_{\omega_i}(o_i^{t'+1}, \boldsymbol{\xi}_i^{t'+1}) - V_{\omega_i}(o_i^{t'}, \boldsymbol{\xi}_i^{t'})) + r_i^T$ denotes the generalized advantage estimation where $\lambda \in [0,1]$ balances variance and bias in value estimation. $V_{\omega_i}$ is the value function that predicts the return from the agent's current information state. Its parameters are learned by minimizing

$$\mathcal{L}_{\text{val}}(\omega_i) = \mathbb{E}\left[\sum_{t=0}^{T}\left(V_{\omega_i}(o_i^t, \boldsymbol{\xi}_i^t) - \sum_{t'=t}^{T}\gamma^{t'-t}r_i^t\right)^2\right]. \tag{3}$$

The second term in Equation 1 is designed to regularize $\psi$ toward neighborhood consensus weighted by $\lambda_{\text{conf}} \geq 0$, capturing the human tendency to consider peers' points of view Pan et al. (2024). Intuitively, if a group of agents shares similar evaluation criteria, their gossip becomes more informative and consistent. Thus, this alignment helps stabilize norm emergence. For agent $i$,

$$\mathcal{L}_{\text{conf}}(\theta_i) = \mathbb{E}\left[\left\|\psi_{\theta_i}(\boldsymbol{\xi}_i^t, \boldsymbol{\xi}_{\mathcal{N}_i}^t) - \frac{1}{|\mathcal{N}_i|}\sum_{j\in\mathcal{N}_i}\boldsymbol{\xi}_j^t\right\|_2^2\right]. \tag{4}$$

As shown in Equation 4, $\mathcal{L}_{\text{conf}}$ measures the distance between agent $i$'s assessments and its neighbors' average assessments for others. This term can be viewed as a graph-based smoothness prior over assessments, improving sample efficiency in sparse interactions without collapsing minority interaction evidence, since $\phi$ can override consensus using fresh interaction data.

The entropy loss $\mathcal{L}_{\text{ent}}(\theta_i) = -\mathbb{E}\left[\sum_{t=0}^{T}\sum_{a\in\mathcal{A}}\pi_{\theta_i}(a|o_i^t,\cdot)\log\pi_{\theta_i}(a|o_i^t,\cdot)\right]$ is added to encourage exploration (Haarnoja et al., 2017). We leave the pseudocode of COOPER in Appendix A.

## 5 EMPIRICAL RESULTS

To comprehensively validate the effectiveness of COOPER, we conduct extensive experiments in both matrix-form and extended mixed-motive games across various social networks. These experiments assess COOPER's capabilities in two key aspects: adaptation to the environment with existing reputation norms, and the co-emergence of reputation norms and cooperation in self-play settings.

### 5.1 ENVIRONMENTAL SETUP

As shown in Figure 1, our environment includes a social network where agents diffuse reputation assessment among neighbors, and game playing where agents are randomly paired to interact and receive rewards. We focus on three classic network structures: small-world, scale-free, and fully connected networks. A detailed introduction to these network structures is provided in Appendix C.

Regarding the game scenarios, we consider the **Donation Game** and the **Coin Game** on a grid world. In the donation game, one agent is the donor and the other is the recipient. The donor can choose to donate, incurring a cost of $c \geq 0$ and giving the recipient a benefit of $b > c$. If not, there is no cost or reward. Coin game is set in a $5 \times 5$ grid world with two agents of different colors. A randomly colored coin is placed randomly. Both agents can pick up the coin and receive a reward of 1. If the coin's color does not match the agent's color, the other agent incurs a penalty of $-2$.

Each experiment has a population size of 10 and an episode length of 50. In each episode, all agents start with 0 reputation for each other. At each step, two agents are randomly paired to play a game. As for the interaction-based assessment update, we apply a "memory-two" setting, where agents rely on the previous two interactions to assign assessments. More concretely, $\mathcal{H}_{i,g^t(i)}^t = [a_{g^t(i)}^t, a_i^t, a_{g^t(i)}^{t_1}, a_i^{t_1}]$ contains agent $i$'s last two interactions with co-player $g^t(i)$.

**Baselines:** We compare our agent with **PPO** (Schulman et al., 2017) and existing reputation-based reinforcement learning agents. **LR2** (Ren et al., 2025) builds on PPO, using reputation as an intrinsic reward with $r_{\text{total}} = [\beta + (1-\beta)\xi_{\mathcal{N}_i \to i}] \times r_i$, where $\beta = 0.5$ and $\xi_{\mathcal{N}_i \to i}$ is the average assessment of $i$ assigned by its neighbors. **RR** (Smit & Santos, 2024) uses *Stern Judging*, a predefined reputation

assessment rule, to update reputation. **IR** (Anastassacos et al., 2021) relies on rule-based agents to foster the learning of reputation assignment rules and the reward is designed as $r_{\text{total}} = \alpha r_i + (1 - \alpha)S_i$, where $S_i$ is calculated assuming the co-player uses a same strategy as $i$.

## 5.2 ADAPTATION

This subsection tests COOPER's adaptability from three aspects: i) whether COOPER can recognize and adapt to unknown reputation norms and reputation-based policies? ii) whether COOPER can distinguish between different norms and reputation-based policies? iii) whether COOPER can leverage the reputation mechanism to promote cooperation? To answer these questions, we introduce one COOPER agent into an environment with pre-defined reputation norms and policies.

We augment three classic rule-based agents with reputation awareness (RA). The ALLC-RA agent cooperates if the co-player's reputation is above a threshold (e.g., -0.5) and defects otherwise. The ALLD-RA agent defects by default but cooperates if the co-player's reputation exceeds a threshold. The TFT-RA agent initially cooperates, then mirrors the co-player's previous action, switching to defection if the co-player's reputation falls below a threshold. All rule-based agents update their reputation assessments based on both social information from neighbors and game interaction.

For gossip-based updates, rule-based agent $i$ computes $\bar{\xi}_i^t$ by the average assessment updates of its neighbors: $\bar{\xi}_i^t = \min\left(1, \max\left(-1, \xi_i^t + \frac{\sum_{j \in \mathcal{N}_i}(\xi_j^t - \bar{\xi}_j^{t-1})}{|\mathcal{N}_i|}\right)\right)$. After interaction, agent $i$ adjusts the assessment by adding or subtracting $\delta = 0.25$ based on whether the co-player cooperates or defects: $\xi_{i \to g^t(i)}^{t+1} = \min\left(1, \max\left(-1, \bar{\xi}_{i \to g^t(i)}^t \pm \delta\right)\right)$.

### 5.2.1 COOPER ADAPTS TO EXISTING REPUTATION NORM

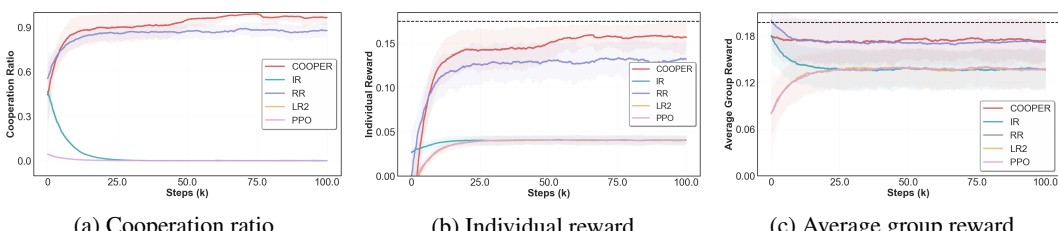

| (a) Cooperation ratio | (b) Individual reward | (c) Average group reward |

Figure 2: COOPER achieves a high cooperation ratio and rewards compared to baselines when adapting to TFT-RA agents. The dashed line denotes the performance upper bound.

The background agents are TFT-RA with a threshold of $0.25$. The social network $G$ is modeled as a small-world network with size $n = 10$, average degree $k = 4$, and the rewiring probability $p = 0.2$. As shown in Figure 2a, COOPER quickly learns to cooperate with the TFT-RA agents. Based on its initial cooperation, COOPER earns a favorable reputation assessment from TFT-RA agents. Once its reputation surpasses TFT-RA's cooperation threshold, TFT-RA stabilizes its own behavior into a cooperative mode. As a result, the interactions between COOPER and TFT-RA become mutual cooperation, which yields high payoffs for both agents as shown in Figure 2b and Figure 2c.

### 5.2.2 COOPER STIMULATES COOPERATION

Here, a COOPER agent is placed into a population of ALLD-RA agents with a threshold of $0.5$. The agents are embedded in a scale-free network with size $n = 10$, neighbor number $m = 2$. An example is shown in Figure 3 where the node size is proportional to its degree.

Although a homogeneous group of ALLD-RA agents (with a threshold of $0.5$) would converge to defection, COOPER learns to break this equilibrium. As shown in Figure 4a, COOPER identifies the reputation-based strategy of ALLD-RA and sustains a high cooperation rate to meet their threshold, incentivizing them to switch to cooperation. This results in sustained cooperation, which is directly reflected in its high individual reward in Figure 4b.

Figure 3: Scale-free network.

In addition, when COOPER is positioned at the hub of a scale-free network, its high reputation and cooperative behavior are rapidly disseminated to the entire population via gossip. More importantly, after mutual cooperation, ALLD-RA is assigned a higher assessment by COOPER. This creates a positive feedback loop: the improved reputation of ALLD-RA agents encourages cooperation not only with COOPER but also among themselves. Consequently, COOPER acts as a "cooperation seed" that triggers a cascade of cooperation, leading the entire group to achieve a higher average reward compared to baseline methods, as shown in Figure 4c.

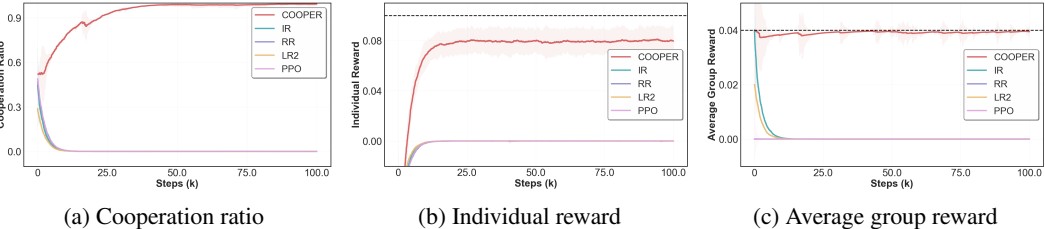

(a) Cooperation ratio    (b) Individual reward    (c) Average group reward

Figure 4: COOPER achieves a high reward compared to baselines and stimulates cooperation in ALLD-RA crowds. The dashed line denotes the performance upper bound.

### 5.2.3 COOPER IDENTIFIES DIFFERENT CO-PLAYERS AND PROMOTES COOPERATION

The background population consists of 2 ALLC agents without reputation sensitivity, 2 ALLD-RA agents with a threshold of $0.5$, and 5 TFT-RA agents with a threshold of $0.25$. The social network $G$ is modeled as a scale-free network with network size $n = 10$, neighbor number $m = 2$.

In Figure 5b, COOPER sustains a high cooperation rate toward both ALLD-RA and TFT-RA to satisfy their thresholds, while slightly lowering its cooperation toward ALLC to exploit their unconditional cooperation. COOPER also recognizes that reputation propagates through the network and fully defecting ALLC will negatively affect its reputation. In Figure 5a, we can see that COOPER's reputation management fosters widespread mutual cooperation, resulting in a higher average group reward. Furthermore, the results shown in Figure 5c indicate that COOPER learns to assign distinct reputations to the three types of agents, and thus can adopt different strategies accordingly, demonstrating its robustness in learning and adapting to heterogeneous reputation norms.

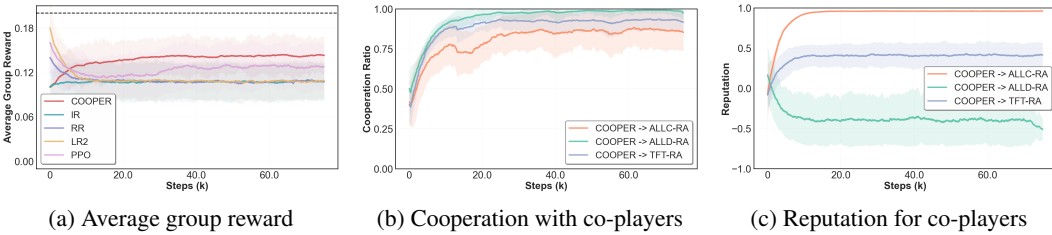

(a) Average group reward    (b) Cooperation with co-players    (c) Reputation for co-players

Figure 5: COOPER identifies different co-players and achieves a high reward compared to baselines.

### 5.3 SELF-PLAY

This subsection examines: i) whether COOPER can establish reputation-based cooperation from scratch; ii) what the emerging reputation norm is like; and iii) how the two reputation-assignment modules $\psi$ and $\phi$ contribute to COOPER's performance. Given that RR and IR require predefined reputation update rules, it would be inappropriate to compare COOPER's self-play performance with these methods. Instead, we compare COOPER's performance with PPO and LR2.

Figure 6a demonstrates COOPER's capability to establish reputation-based cooperation across diverse network topologies. Compared to baselines, COOPER consistently performs the best across various social networks (additional results are shown in Appendix E). Specifically, networks with higher degree heterogeneity, i.e., the scale-free network, support the highest cooperation ratios, followed by small-world and then fully connected networks. This pattern aligns with established evolutionary game theory literature, which suggests that heterogeneous network structures can promote cooperation through hubs acting as cooperation anchors (Santos & Pacheco, 2005; Perc et al., 2017).

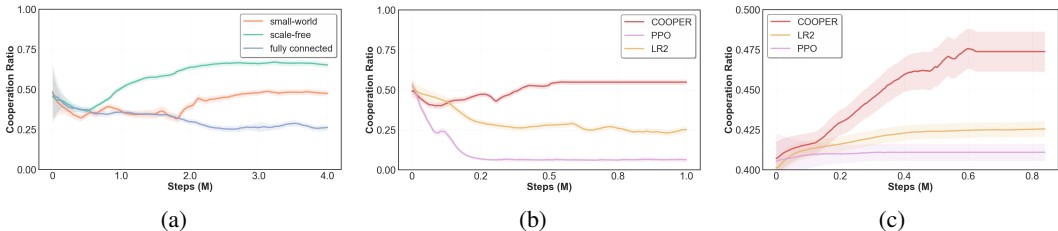

(a)                                      (b)                                      (c)

Figure 6: COOPER achieves high cooperation ratio in self-play. (a) shows the performance of COOPER in donation games $b = 0.5, c = 0.3$ with various social networks. (b) shows the cooperation ratio compared with baselines in the donation game with $b = 0.5, c = 0.1$ on fully connected networks. (c) depicts the performances in the grid world coin game on a scale-free network.

Notably, we employ a moderated social dilemma setting where cooperation costs $c = 0.1$ and benefits the recipient $b = 0.5$ for fair comparison with LR2, as LR2 struggles with more extreme dilemmas (we use $b = 0.5, c = 0.3$ in other experiments). Figure 6b shows the performance of the three approaches in fully connected networks. The results show that COOPER outperforms the baselines, while the PPO agents fail to establish cooperation and converge to defection.

Figure 6c illustrates the performance of the algorithms in the grid world coin game with population size $n = 10$. Agents are embedded on a scale-free network with average neighbor number $m = 2$. In this more complex environment, COOPER still exhibits cooperative behavior, though the improvement is moderate. This indicates that COOPER learns to cooperate using reputation information, but additional context should be provided to further enhance cooperation. Future work could explore incorporating more historical or reputational information to facilitate agents' decision-making.

We next analyze the fundamental reasons behind COOPER's superior performance compared to the baselines. Consider the scenario where an agent cooperates but its co-player defects, which is common during the early stages of norm formation. In this case, the agent receives an environment reward $r_i = -c$. Under LR2's reward formulation, $r_{\text{total}} = \beta \cdot r_i + (1 - \beta) \cdot \xi_{\mathcal{N}_i \rightarrow i} \cdot r_i$. For an agent with higher reputation ($\xi_{\mathcal{N}_i \rightarrow i} \rightarrow 1$), $r_{\text{total}}$ is closer to $r_i$, meaning that the negative environmental reward is **amplified** (positive rewards is also scaled, but the negative ones create disincentive to cooperate.), creating a perverse incentive where *higher reputation leads to greater punishment for unilateral cooperation*. This design flaw provides **misaligned learning signals** that discourage cooperative behavior in challenging scenarios. COOPER avoids this pitfall by jointly learning a reputation assignment module and a reputation-based policy without relying on extra reputational rewards. It is also worth noting that the PPO agent, lacking reputation modules, fails to develop farsighted strategies and sustained cooperation in these mixed-motive environments.

### 5.3.1 EMERGED NORM

In a fully connected network with $n = 10$, all agents converge to the same reputation norm, whereas in a scale-free network with population size $n = 10$ and average neighbor $m = 2$ shown in Figure 3, hub and leaf agents develop different behavior patterns. Since reputation assignment modules and reputation-based policy are learned distributively, the interpretation of reputation values can vary among agents (e.g., one may regard $\xi_{i \rightarrow j} = 0.7$ as favorable, another may not). To quantify and compare agents' reputation norms, we visualize how agents update and subsequently utilize reputation in decision-making using an experience-to-action heatmap. The x-axis represents the most recent joint action, where 'CC' denotes mutual cooperation, 'CD' indicates that the opponent cooperated while the focal agent defected, and so on for 'DC' and 'DD'. The y-axis shows the previous interaction. Each cell indicates the probability of cooperation in the next encounter with the same opponent, given the past two interactions. For example, in Figure 7, the upper right cell shows the agent will cooperate if the previous sequence was $[D, D, C, C]$.

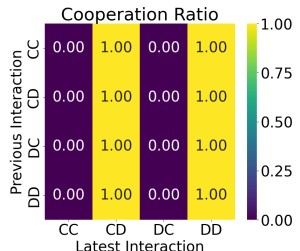

Figure 7: Norm example.

Take agent 1's reputation-based cooperation pattern shown in Figure 7 as an example (see Appendix F for other agents' emerged behavioral pattern). In fully connected networks, agents tend to defect against partners with whom they previously cooperated, while cooperating with those they

previously defected. In scale-free networks, a more complex norm emerges. Hub agent, as presented in Figure 8a, with their numerous connections, develop strategies similar to those in the fully connected networks. In contrast, leaf agents with limited connections exhibit a more nuanced pattern shown in Figure 8b: they always cooperate unless mutual cooperation (CC) in the latest interaction, in which case they cooperate probabilistically. The diverged behavioral norms can be explained by agents' structural positions. Hub agents, owing to their high connectivity, are exposed to multiple information sources in gossip that approximate a well-mixed environment. Leaf agents with limited connections, on the other hand, depend heavily on localized information and direct experience. This constraint leads them to adopt more cautious and generally more cooperative strategies.

The "leading eight" norms proposed by Ohtsuki & Iwasa (2006), have been pivotal in reputation study, providing a foundational framework for understanding how simple rules can drive cooperative behavior. We plot the "leading eight" norms with the initial self-assessment $\xi_{i \to i}$ set to 1 in Figure 8c for comparison. The rule-based reputation norms generally play cooperation if the co-player cooperated in the latest interaction and defection otherwise. In contrast, our method can develop flexible reputation norms that are related to the network structure, which further leverages reputation norms as well as the network structural features to promote cooperation.

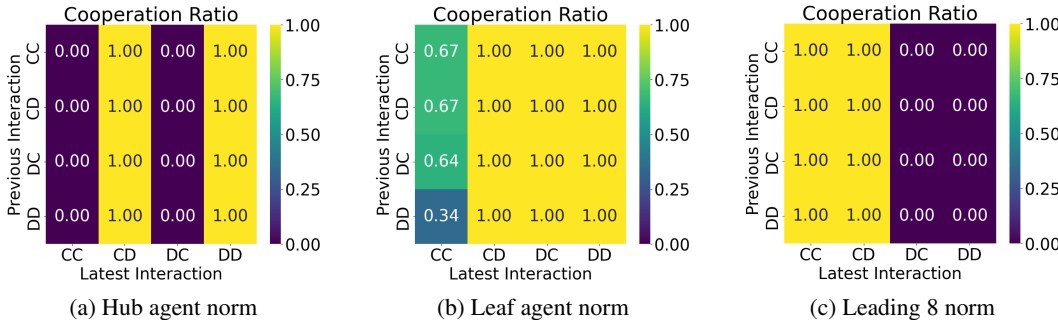

(a) Hub agent norm   (b) Leaf agent norm   (c) Leading 8 norm

Figure 8: Hub agent and leaf agent in the scale-free network learn different patterns. (c) presents the "leading eight" norms where the initial self-assessment $\xi_{i \to i}$ is 1[1].

### 5.3.2 ABLATION STUDY

In this section, we conduct an ablation study in a 10-agent donation game self-play setting on a scale-free network with $m = 2$. Specifically, we evaluate two ablated versions of COOPER: 1) COOPER without $\psi$, which lacks the gossip-based reputation assessment and relies solely on interaction experiences, and 2) COOPER without $\phi$, which removes the interaction-based assessment module and depends exclusively on social gossip.

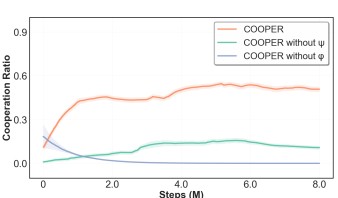

Figure 9: Ablation study

In Figure 9, the removal of either module leads to a noticeable decline in cooperation ratio. The absence of the gossip module $\psi$ results in a significant drop in cooperation, a finding consistent with theoretical work on the co-evolutionary relationship between gossip and reputation-based cooperation (Pan et al., 2024). This decline occurs because, without social information sharing, agents are limited to their own interactions, thereby slowing the spread of reputation assessment and hindering reputation-based cooperation. When the interaction-based module $\phi$ is removed, cooperation fails to emerge entirely. The reason is that without interaction-based assessments, reputation updates remain unanchored and fail to provide reliable guidance for action.

## 6 CONCLUSION

We propose **COOPER**, a reinforcement learning algorithm that jointly learns reputation assignment modules and policies without pre-defined rules or additional reward shaping. Extensive experiments show that COOPER can adapt to existing norms and develop emergent reputation norms to promote cooperation in decentralized multi-agent systems. See Appendix H for discussion of future work.

---

[1]To plot the leading eight heatmap, the initial self-assessment is either Good $\xi_{i \to i} = 1$ or Bad $\xi_{i \to i} = -1$.

## ETHICS STATEMENT

We confirm that our research, which focuses on designing a novel MARL algorithm and evaluating within a simulated environment, does not raise any ethical concerns. This work does not involve human subjects, personal data, or real-world deployments. It therefore poses no risks to privacy, safety, or well-being. We have designed the study in accordance with principles of scientific rigor, transparency, and reproducibility, and affirm that it aligns with the ethical guidelines set forth by ICLR.

## REPRODUCIBILITY STATEMENT

We have included detailed descriptions of our method and experimental setup in the main text and appendix to facilitate reproducibility. Section 4, along with Appendices A and B, elaborates on our proposed algorithm and implementation specifics, including critical hyperparameters and computational resources utilized for training. As for the experimental environment setting, we thoroughly presented the background population's setup and reward structures in Section 5.1 and Appendix D. In our experiments, all reported results are averaged over 6 independent runs with different random seeds, and the corresponding standard deviations are provided. We plan to release the full source code, along with configuration files and scripts for reproducing all experiments, upon publication.

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

# A  ALGORITHM

---

**Algorithm 1 COOPER** (**COOP**eration with **E**mergent **R**eputation)

---

Initialize policy network $\pi_i$ with parameters $\theta_i$, value function $V_i$ with parameters $\omega_i$, gossip module $\psi_i$, interaction module $\phi_i$; replay buffer $\mathcal{B}_i \leftarrow \emptyset$ for each agent $i \in N$.
Initialize all reputations $\xi_{i \rightarrow j} \leftarrow 0$, $\forall i, j \in N$.
**for** episode $= 1$ to $T$ **do**
    Reset environment, get initial observations $o_i^0$ and initial reputations $\boldsymbol{\xi}_i^0$ for all agents.
    **for** timestep $t = 0$ to $T_{\max} - 1$ **do**
        **for** each agent $i$ **do**
            $\bar{\boldsymbol{\xi}}_i^t \leftarrow \psi_i \left( \boldsymbol{\xi}_i^t, \boldsymbol{\xi}_{\mathcal{N}_i}^t \right)$                             $\triangleright$ Gossip-based update
            $a_i^t \sim \pi_i \left( o_i^t, \bar{\boldsymbol{\xi}}_i^t \right)$             $\triangleright$ Sample action (with $\xi_{i \rightarrow g^t(i)}$ and $\xi_{N \rightarrow i}$)
        **end for**
        Execute joint action $\boldsymbol{a}^t$, observe rewards $\boldsymbol{r}^t$, next observations $\boldsymbol{o}^{t+1}$.
        **for** each agent $i$ **do**
            $\boldsymbol{\xi}_i^{t+1} \leftarrow \phi_i \left( \bar{\boldsymbol{\xi}}_i^t, \mathcal{H}_{i,g^t(i)}^t \right)$                  $\triangleright$ Interaction-based update
            Store $\left( o_i^t, \boldsymbol{\xi}_i^t, a_i^t, r_i^t, o_i^{t+1}, \boldsymbol{\xi}_i^{t+1} \right)$ in $\mathcal{B}_i$
        **end for**
    **end for**
    **Update Phase:**
    **for** each agent $i$ **do**
        Sample minibatch $\mathcal{M}_i$ from $\mathcal{B}_i$
        Compute generalized advantage estimates $\hat{A}_i^t$ using $V_{\omega_i}$.
        Compute $\mathcal{L}_{\text{env}}$, conformity loss $\mathcal{L}_{\text{conf}}$, and entropy regularization $\mathcal{L}_{\text{ent}}$. Then, update $\theta_i$ by minimizing $\mathcal{L}(\theta_i) = \mathcal{L}_{\text{env}} + \lambda_{\text{ent}}\mathcal{L}_{\text{ent}} + \lambda_{\text{conf}}\mathcal{L}_{\text{conf}}$.
        Update $\omega_i$ by minimizing $\mathcal{L}_{\text{val}}$.
    **end for**
**end for**

---

# B  IMPLEMENTATION DETAILS

Our agent's policy module is implemented based on PPO algorithm Schulman et al. (2017). We adopt Adam optimizer for all modules training. The assessment module $\phi$ updates the reputation based on the action sequence and current reputation. It takes these inputs, concatenates them, and passes them through a series of fully connected layers with Tanh activations to produce an updated reputation value. The communication module $\psi$ processes the combined representation of neighbor reputation and the agent's own reputation. It consists of two separate processing branches for neighbor and RL reputation, respectively. These branches are concatenated and passed through additional layers to produce an output vector representing the integrated reputation information. In coin game, we add an observation processor that processes the coin game observation using convolutional layers. The input observation is first permuted to match the channel-first format required by convolutional layers. The network consists of two convolutional layers followed by flattening and fully connected layers, ultimately producing a processed observation that matches the original observation size.

**Parameter Design**

As for parameter design, the learning rate for the optimizer, set to 2.5e-4. The discount factor for the reward, set to 0.99. The lambda value for Generalized Advantage Estimation (GAE), set to 0.95. The clipping coefficient for the PPO (Proximal Policy Optimization) algorithm, set to 0.3. The entropy coefficient, set to 0.05. This parameter encourages exploration by adding an entropy term to the loss function.The value function coefficient, set to 0.5.The maximum norm for gradient clipping, set to 0.5.The number of mini-batches used for updating the policy, set to 4.

In practice, we tune $\lambda_{conf}$ empirically. In self-play settings, where agents learn from scratch, conformity is less critical, so we set $\lambda_{conf} = 0$. In adaptation settings, where agents interact with rule-based agents with existing reputation norms, we set $\lambda_{conf} = 0.5$ to encourage alignment with

the existing norms. For a given environment, the best coefficient value can be selected via grid search over simulations. As for the rule-based reputation update introduced in Section 5.2, the parameter $\delta = 0.25$ is chosen empirically as a moderate step size that allows meaningful but not overwhelming reputation adjustments per interaction (given that the reputation is ranging from -1 to 1).

**Experiments Computer Resources** CPU:13th Gen Intel(R) Core(TM) i9-13900KF;Total memory:64.0 GB;GPU:NVIDIA GeForce RTX 4090;Memory per GPU:55.9 GB.

## C  SOCIAL NETWORKS

In our model, reputation information diffuses on a social network and influences agents' reputation assessment. To investigate how different social network structures influence the emergence of cooperative behavior and the effectiveness of reputation mechanisms, we conduct experiments across various classic network structures.

**Scale-Free networks** are characterized by a power-law degree distribution, where a few nodes (hubs) have a significantly higher number of connections compared to the majority of nodes. This structure mimics real-world social networks, where a small number of highly connected individuals play a crucial role in information spread and influence.

In our experiments, we generate scale-free networks using the Barabási–Albert model, which incorporates preferential attachment. This mechanism leads to the formation of hubs, where new nodes are more likely to connect to nodes that already have a high number of connections. For a scale-free network parameterized with $n = 10$ and $m = 2$, the network consists of $n = 10$ agents. The network is constructed through a process where each new node, as it is added to the network, forms $m = 2$ connections to existing nodes (if possible). The probability of forming a connection to an existing node is proportional to the number of connections that the existing node already has. This preferential attachment mechanism ensures that nodes with more connections are more likely to attract additional connections as the network grows.

**Small-World networks** combine high clustering with short average path lengths, which makes them efficient for information dissemination while maintaining local connectivity. These networks are generated using the Watts–Strogatz model, which starts with a regular lattice and rewires some edges with a certain probability to introduce randomness while preserving local structure. Small-world networks are useful for studying how local interactions and global connectivity impact agent behavior.

Specifically, in our experiments, the parameters are $n = 10, k = 4, p = 0.2$, meaning that each agent is initially connected to its four nearest neighbors in a ring, forming a regular lattice. Every existing edge is then rewired independently with a probability of 0.2 to create long-range ties. This process introduces shortcuts that reduce the average path length while maintaining a high clustering coefficient, thus capturing the essence of small-world properties.

**Fully connected networks** (also known as complete graphs) are networks where every node is directly connected to every other node, creating a well-mixed structure where all nodes are equivalent in the gossip phase, i.e., all agents have the same social neighbor set. Moreover, the well-mixed nature of fully connected networks allows us to isolate the effects of network topology, providing a clearer picture of how norms and behaviors spread and stabilize in a homogeneous environment.

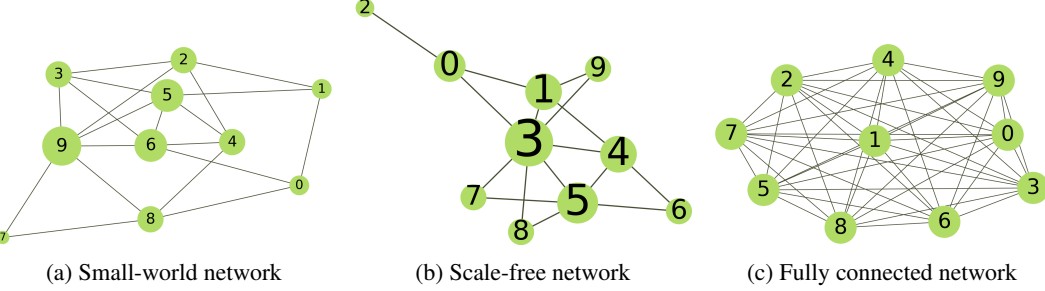

(a) Small-world network    (b) Scale-free network    (c) Fully connected network

Figure 10: Examples of small-world, scale-free, and fully connected networks.

## D    DETAILED ENVIRONMENTAL SETUP

In our model, agents are embedded on a social network, where reputation information propagates through the underlying graph structure and influences how agents update their assessments. To examine the robustness of our method, we conduct extensive experiments across several classic network structures. A detailed introduction to the network structures used in our study is provided in Appendix C. In each experiment, the population size is 10, and the episode length is 50. In each episode, all agents are initialized with 0 reputation for each other. At each step, every two agents are randomly paired up to play a game. Regarding the limited memory capacity of agents with respect to interaction history, we apply a "memory-two" setting without loss of generality. In particular, $\mathcal{H}_{i,g^t(i)}^t = [a_{g^t(i)}^t, a_i^t, a_{g^t(i)}^{t_1}, a_i^{t_1}]$ contains agent $i$'s last two interaction history with co-player $g^t(i)$, i.e., $t_1$ is the last time agent $i$ and $g^t(i)$ interacted before $t$.

**Donation Game** is an extensively studied game in the field of reputation and evolutionary game theory. In this pairwise game, one agent acts as the donor and the other as the recipient. The donor has the option to either donate or not. If the donor chooses to donate, they incur a cost of $0.3$, while the recipient receives a benefit of $0.5$. Conversely, if the donor does not donate, there is no cost to the donor, and the recipient receives no reward. This setup mirrors real-world cooperative behaviors, such as giving away second-hand goods, helping others.

**Coin Game** is set in a grid world, involving two agents with different colors interacting on a $5 \times 5$ map where a randomly colored coin is randomly placed. Both agents can pick up the coin and receive a reward of $1$. After a coin is collected, a new coin is generated at a random location. However, if the color of the coin does not match the color of the agent picking it up, the other agent incurs a penalty of $-2$. This dynamic creates a sequential social dilemma. While each agent may be tempted to collect any coin they encounter, doing so can result in a lower overall expected reward. Therefore, agents need to learn to refrain from collecting coins that do not match their own color, allowing their co-player to do so instead, in order to maximize their long-term rewards.

We compare the performance of our agent against the **PPO** (Schulman et al., 2017) agent without a reputation module as well as other existing reputation-based reinforcement learning agents.

**LR2** (Ren et al., 2025) is built on PPO and it takes reputation as an intrinsic reward and define the reward function as $r_{\text{total}} = [\beta + (1 - \beta)\, \xi_{N \to i}] \times r_i$, where $\beta \in [0, 1]$ denotes how much one values a higher reputation. We set $\beta = 0.5$ in the experiments as in the original paper. In this approach, one's reputation $\xi_{N \to i}$ is directly computed by the average assessment assigned by neighbors: $\xi_{N \to i} = \frac{\sum_{j \in \mathcal{N}_i} \xi_{j \to i}}{|\mathcal{N}_i|}$.

**RR** (Smit & Santos, 2024) follows a pre-defined reputation assignment rule to update reputation for others, and the agents take actions conditioning on the co-player's reputation with classic reinforcement learning algorithms. In the experiment, we implement Stern Judging, one of the extensively studied "leading eight" norms (Ohtsuki & Iwasa, 2006), which only assigns a bad reputation if agents with a higher (lower) reputation cooperate with one with a lower (higher) reputation.

**IR** (Anastassacos et al., 2021) adds seeding agents with existing reputation rules to foster reputation norm in group and introduce an introspective reward to promote reputation-based cooperation, $r_{\text{total}} = \alpha r_i + (1 - \alpha)S_i$ where $\alpha$ balances the weight of actual reward and imaginary reward. The imaginary reward $S_i$ is generated by assuming $i$'s co-player plays the same strategy as $i$. In the experiment, we apply a straightforward implementation and provide the agents with a game matrix so that they can calculate the introspective reward.

To ensure fair comparison, every baseline agent receives exactly the same raw observation as a COOPER agent. Hence, there is no difference in the MDP state/observation between methods. In baselines without reputation-related modules, such as PPO, reputation information is treated as just one more entry in the observation vector. Additionally, we want to clarify that our PPO baseline is not "vanilla" in the sense of lacking regularization. We keep the same actor–critic network width, entropy coefficient, clipping coefficient, learning rate, and so on, as COOPER; the only change is that the reputation assignment modules and the reputation-related loss are removed. We follow CleanRL's PPO implementation and ensures that any performance gap comes solely from the reputation related learning framework and module design.

# E    SELF-PLAY DONATION GAME ON DIFFERENT NETWORKS

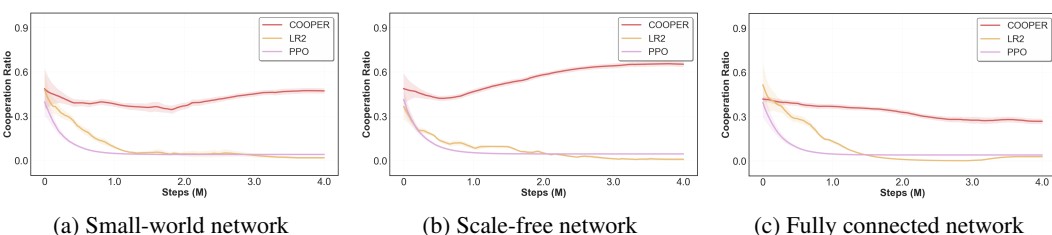

(a) Small-world network            (b) Scale-free network            (c) Fully connected network

Figure 11: Self-play in donation game $b = 0.5, c = 0.3$ on various networks with network size $n = 10$.

Following the standard MARL convention, self-play refers to a setting where all agents are COOPER agents, and they jointly learn from scratch without any pre-defined reputation rules or external supervision.

We conduct self-play experiments in the donation game, with $b = 0.5, c = 0.3$, on three classic network structures. Figure 11a is conducted on small-world networks with average neighbor number $k = 4$. Figure 11b shows the performance on scale-free networks with $m = 2$. Figure 11c presents the emerged cooperation on fully connected networks. In this rather strict (compared to the donation game with $b = 0.5, c = 0.1$) social dilemma setting, COOPER outperforms the baselines despite the social network structure.

## E.1    NETWORK SIZE AND SCALABILITY

We repeat the self-play donation-game experiment with population sizes n = 10, 30, 60, while keeping the other network parameters the same. Due to the population changes, we correspondingly update the episode length to 50, 150, 300 steps (so that every two agents will have approximately 5 encounters in each episode). As shown in Figure 12, the results of the self-play experiment confirm that COOPER achieves a cooperation level that outperforms the baselines across all population sizes, but the cooperation ratio is lower for large populations given the same sampling steps. To be more specific, with 30k steps, in scale-free networks, n=30 achieves an average cooperation ratio of 0.398, and n=60 achieves an average cooperation ratio of 0.336 (n=10 achieves an average cooperation ratio of 0.726).

We speculate that the decline in cooperation is tied to gossip-efficiency differences across scales. In the 10-agent scale-free network generated by the Barabási–Albert model with m = 2, the average path length is only about 1.5-1.7 hops, so a reputation update reaches the whole population almost immediately. With 30 agents (same m = 2), the average path length grows to 2.4-2.6 hops (and for n=60, 2.9-3.1 steps), slowing convergence of the reputation estimates and weakening the signal that COOPER needs to sustain high cooperation levels.

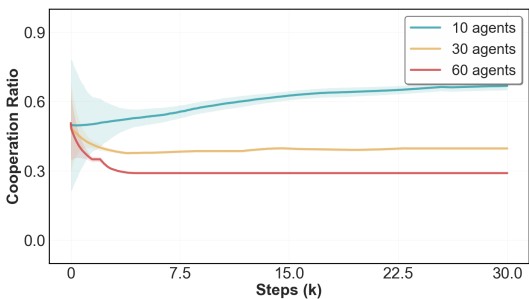

Figure 12: Self-play Cooperation Ratio with 10, 30, 60 agents

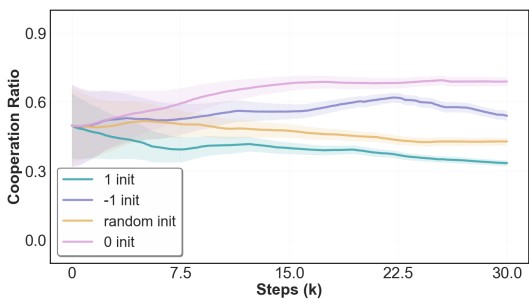

Figure 13: Cooperation Level Difference Caused by Initialization

## E.2 INITIALIZATION

In this paper, the initial reputation is set to 0 for all agents by default. Since the reputation value ranges from -1 to 1, this initialization indicates a neutral and default starting point. As shown in Figure 13, the initialization difference has some impact on the final stable state, in terms of the final cooperation ratio/reward, but it does not affect the pattern that COOPER learns to cooperate with emergent reputation. To be more specific, for self-play experiments on the scale-free networks with 10 agents, initializing the reputation as all 0, all 1, all -1, and uniformly random results in the cooperation ratio of 0.726, 0.38, 0.56, and 0.48, respectively.

Here, initializing $\xi$ as all ones leads to the worst cooperation level. We conjecture that with $\xi$ initialized as +1, the only possible update direction is downward, and once every reputation has been dragged slightly below +1, the population loses the numerical contrast needed to separate "good" from "bad", and cooperation collapses. The fact that all -1 are better than all +1 indicates that there is a trend to assign a higher reputation to cooperators and a a lower reputation to defectors, so starting with all -1 does not hinder the reputation rise for the cooperators. Moreover, initialization with 0 leaves the full range open where both positive and negative updates are feasible, and this symmetry produces the highest cooperation level.

## F ADDITIONAL RESULTS ON THE EMERGED NORM

In Figure 14, we present each agent's reputation-based cooperation pattern that emerged in the self-play donation game on a fully connected network with network size $n = 10$.

For 10-agent self-play in the donation game on scale-free networks ($n = 10, m = 2$), we visualize the interaction-based reputation module $\phi$ of the hub agent and the leaf agent. As illustrated in Figure 15, the hub agent and the leaf agent develop distinct reputation-assignment rules, which can be attributed to their respective positions within the network.

Under the learned reputation norm, COOPER's policy is a Nash Equilibrium: any unilateral deviation breaks the alternation and reduces the deviator's cumulative payoff. We further analyzed the policy under different reputation values. The agent (as shown in Figure 14) cooperates deterministically when the opponent's reputation exceeds –0.3, and defects when it falls below –0.6. Between these thresholds, the probability of cooperation increases monotonically with the opponent's reputation. Importantly, the agent's own reputation also influences its behavior: higher self-reputation increases the tendency to cooperate at any given opponent reputation level.

As for interaction-based reputation updates, cooperation reduces the opponent's reputation by $\approx 1$, while defection increases it by $\approx 1$. We agree that these are not canonical game-theoretic equilibria, but they are emergent, sustainable, and computationally stable under decentralized learning with reputation.

Let's consider a scenario with two players, A and B. In the well-mixed population, the policy learned by COOPER is as follows:

1. $\xi > -0.3$ play C, $\xi < -0.6$ play D, $-0.6 <= \xi <= -0.3$ play C with $p(\xi) = \frac{\xi+0.6}{0.3}$
2. if A plays C, then $\xi_B - 1$, if A plays D, then $\xi_B + 1$ (and same for B)

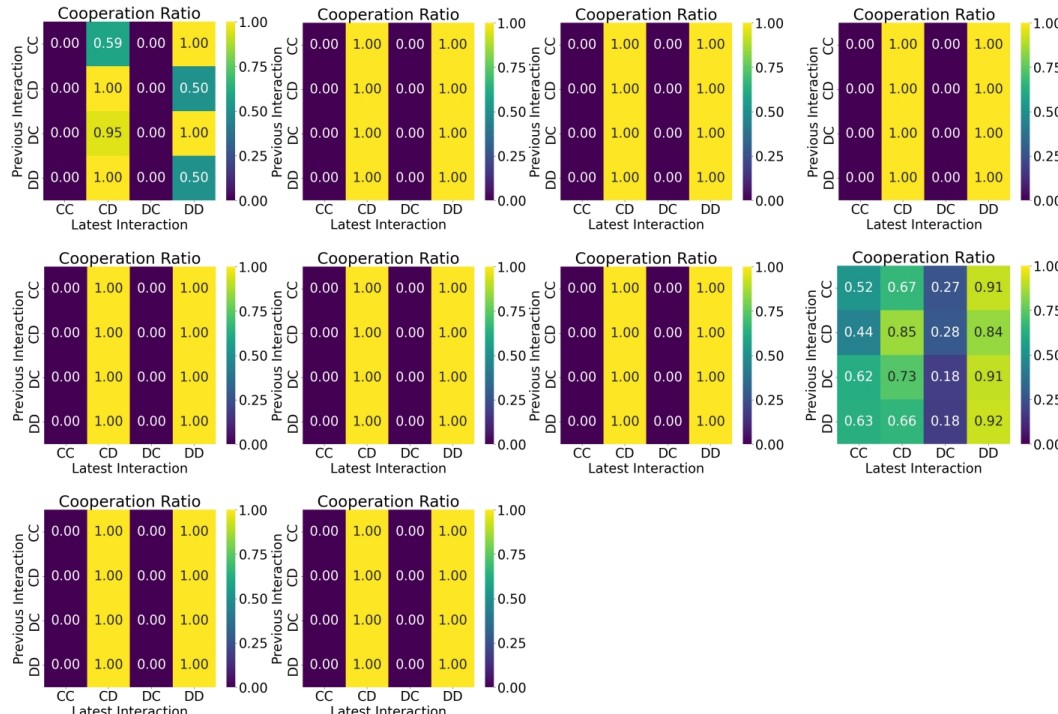

Figure 14: Norm learned in a 10-agent donation game self-play setting. The social network is fully connected. Each heatmap shows the different agents' probability of cooperation given previous interaction.

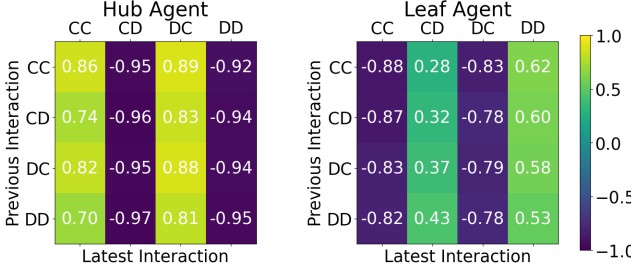

Figure 15: Reputation update pattern of hub and leaf agent in scale-free network. Let the co-player's initial reputation = 0, the heatmap shows the assessment updates under different interaction sequences.

Initially, let $\xi_A = \xi_B = 0$. At step 1, A plays C, B plays C because $\xi_A = \xi_B > -0.3$. So, $r_A = r_B = 0.2$. Then, update $\xi_A = \xi_B = -1$. At step 2, A and B both play D and then update $\xi_A = \xi_B = 0$. And $r_A = r_B = 0$. It is clear that the system is recursively running step 1-2.

But, if at step 1, let agent B deviate and play D, then $r_A = -0.3$, $r_B = 0.5$. In this case, update $\xi_A = 1, \xi_B = -1$. So, in the new step 2, A will play D and B will play C, so $r_A = 0.5$, $r_B = -0.3$. In this case, update $\xi_A = 0, \xi_B = 0$, and it goes back to the situation in step 1.

It is clear that B has no gain in this deviation, so the policy learned by COOPER is stable.

# G  Ablation Study on Different Networks

In this section, we conduct an ablation study in a 10-agent donation game self-play setting on various network structures. The two ablated versions of COOPER are: 1) COOPER without $\psi$, which lacks the gossip-based reputation assessment and relies solely on interaction experiences, and 2) COOPER

without $\phi$, which removes the interaction-based assessment module and depends exclusively on social gossip.

As shown in Figure 16, removing either $\psi$ or $\phi$ leads to a decline in cooperative behavior. Notably, in scale-free networks with apparent degree heterogeneity, the performance drop is more evident when $\psi$ is absent. This highlights the importance of the gossip-based reputation assessment module in maintaining robust cooperation.

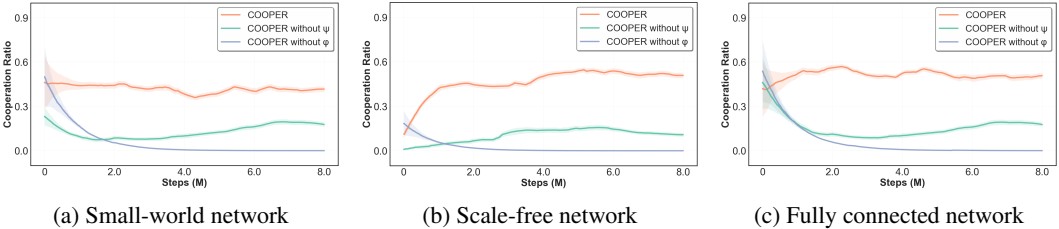

(a) Small-world network  (b) Scale-free network  (c) Fully connected network

Figure 16: Ablation study in different network structures. Both $\psi$ and $\phi$ modules foster cooperation.

### G.1  Additional Comparison with Opponent Shaping Methods

Opponent Shaping (OS) methods are highly relevant to social dilemmas and have shown promise in learning robust strategies. In this subsection, we conduct an additional experiment in the self-play donation game comparing COOPER with two prominent OS methods: Learning with Opponent-Learning Awareness (LOLA) Foerster et al. (2017) and Advantage Alignment Algorithms (AAA) Duque et al. (2024).

As illustrated in Figure 17, COOPER significantly outperforms opponent shaping baselines. A likely explanation is that these baselines struggle to effectively utilize reputation information, which constitutes a substantial portion of the observation space. In contrast, COOPER integrates a gossip-based assignment model ($\psi$) and an interaction-based assignment model ($\phi$), enabling more efficient and accurate processing of reputation-related cues. This dual-model design allows COOPER to better capture and leverage social dynamics, leading to superior performance in self-play settings.

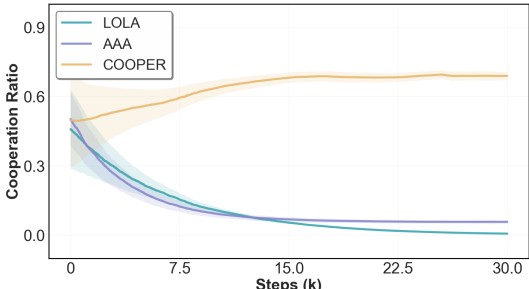

Figure 17: COOPER outperforms the opponent shaping baselines in 10 agents self-play.

## H  Limitation and Future Directions

Despite its promising results, our approach has several limitations that point to important directions for future work. First, the efficacy of COOPER is inherently tied to reliable communication. The model assumes that agents truthfully share their reputation assessments. This is an assumption that may not hold in adversarial settings where agents could disseminate misinformation to manipulate others' reputations for their own benefit. In addition, the current reputation representation, while effective, is a simple scalar value. This may lack the expressiveness to capture complex behavioral nuances in more sophisticated environments, potentially leading to oversimplified or unfair social evaluations. Finally, our study primarily focuses on static network topologies. The dynamics of reputation propagation and norm emergence in dynamically evolving networks, where connections between agents change over time, remain an open and challenging problem.

Assuming only truthful communication during the gossip phase ($\psi$) is a limitation in real-world deployments, as reputation systems are inherently vulnerable to manipulation.

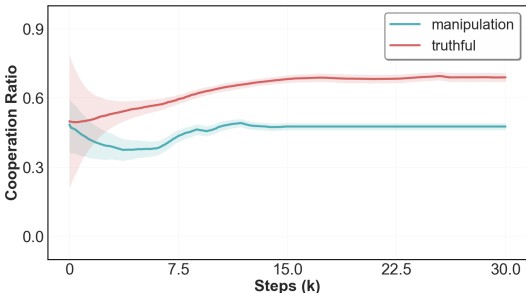

Figure 18: Strategic gossip hinders cooperation in 10 agents self-play Donation Game.

We have conducted preliminary experiments where agents are allowed to manipulate the reputation information they share. Our initial findings suggest that such misinformation can indeed distort group-level cooperation dynamics, reducing group cooperation. As shown in Figure 18, in 10-agent self-play donation game experiments under scale-free networks, the cooperation level with strategic gossip is decreased to 0.47 (while in truthful reputation dissemination, the cooperation level is 0.726).

To mitigate the impact of deceptive gossip on collective cooperation, we can employ adversarial training within COOPER to help agents learn to detect and discount false reputation messages, or simultaneously maintaining an explicit trust score for each neighbor that is updated based on how well their past gossip aligns with subsequent observations. These methods provide a straightforward starting point, though more sophisticated mechanisms may be investigated in future exploration.

