# OpenReview forum: "Learning to Cooperate with Emergent Reputation via Multi-agent Reinforcement Learning"
_ICLR.cc/2026/Conference — ICLR 2026 Conference Withdrawn Submission_

### Official Review · Reviewer_9icq · 2025-10-23

**Soundness:** 1
**Presentation:** 2
**Contribution:** 2
**Rating:** 2
**Confidence:** 5

**Summary:**

This paper considers the problem of learning reputation mechanisms as a way to promote cooperation in multi-agent systems. The authors propose a technique that receives messages from neighbours (as defined by a graph topology), aggregates them to arrive at a reputation assessment, and uses the reputation assessment as input to the agent policy alongside the standard agent observations. The authors carry out evaluations showing that the technique promotes cooperation, can play alongside existing reputation norms, and also leads to cooperation when playing against itself.

**Strengths:**

S1. The work tackles an intelectually interesting problem, is well-grounded in the literature, and aims for a worthwhile goal (designing reputation mechanisms that are highly flexible).

**Weaknesses:**

W1. The approach assumes a single, static graph which is highly limiting.

W2. The evaluation leaves much to be desired and, in my view, does not support the overwhelmingly positive narrative in the paper.

**Questions:**

C1. Regarding the methodology, in my view, the single, static graph assumption is highly limiting. The work cannot account for the agent population growing (e.g., new nodes joining the network). It also cannot generalise across different topologies. Both these aspects may be addressed by using graph neural networks (GNNs) for performing function approximation.

C2. The evaluation is very small-scale and, in my opinion,  there is insufficient evidence to support the claims.
- 4 runs is much too little to draw reliable conclusions in RL; see, for example [1] and [2]
- The experiments are carried out over few very small (n=10) graph instances.  The work should demonstrate the findings hold over substantially more instances (consider pairing a run seed with a graph generation seed as an alternative experimental design) and over multiple numbers of agents.
- A sensitivity analysis should be performed over the network generation parameters; otherwise the values selected seem completely arbitrary

C3. The method is compared with vanilla PPO. But why? COOPER itself is a maximum entropy actor-critic architecture, with the reputation module added in. PPO is much simpler, and the performance differences may well be explained by the difference in architecture. COOPER should be compared with the same RL approach but without the reputation module and associated losses.

### Smaller comments

C4. For the experiment in Section 5.3, could you clarify the setup? Are all agents COOPER agents?

C5. Typos etc: "collectively" -> "collective" (L39), "Individuals help others" -> Add e.g. "The fact that" at the start (L86)

C6. It's probably cleaner to present all loss components from equation 1 and only introduce the one in equation 3 after the others (since it's the loss for the critic)

C7. You should clarify what Markov Decision Processes the baselines operate with (or specify only their differences, e.g., how do observations differ?)

### References

[1] Henderson, P., Islam, R., Bachman, P., Pineau, J., Precup, D., & Meger, D. (2018, April). Deep reinforcement learning that matters. In Proceedings of the AAAI conference on artificial intelligence (Vol. 32, No. 1).

[2] Agarwal, R., Schwarzer, M., Castro, P. S., Courville, A. C., & Bellemare, M. (2021). Deep reinforcement learning at the edge of the statistical precipice. Advances in neural information processing systems, 34, 29304-29320.

---

> ### Author Response · Authors · 2025-11-26
>
> Thank you for your thorough review and constructive feedback. We appreciate your recognition of our work's intellectual merits and grounding in literature. Please find our point-by-point responses below.
>
> **Regarding your suggestion on GNN:**
>
> We appreciate the suggestion. In the current setting, each agent only receives the reputation vectors of its one-hop neighbours, whose number is bounded by the maximum degree. There is no need to process the entire graph, so processing information received from neighbors alone is sufficient. Thus, introducing a GNN would therefore add computational cost without much performance gains. For this reason, we leave the present architecture as-is.
>
> **Regarding the Self-play setup:**
>
> In our context, self-play refers to a setting where all agents are COOPER agents, and they jointly learn from scratch without any pre-defined reputation rules or external supervision. This is now stated explicitly at the beginning of Section 5.3.
>
> **Regarding the experiment design:**
>
> Thank you for pointing out the possible improvements to make our empirical results more robust and convincing.
> Due to computational and time constraints during the rebuttal, we have increased the number of independent runs from 4 to 6 using different random seeds for all experiments. Also, we want to clarify and emphasize that in our submitted experiments, graph instances were already generated using various random seeds and different established generators (Watts-Strogatz, Barabasi–Albert). All curves report standard-error shaded regions, and COOPER’s advantage remains clear and statistically stable even after accounting for this variance. This ensures our findings are not dependent on a single network topology but are evaluated across a spectrum of standard graph structures. The parameters for these graph generators were not arbitrary; they were selected based on a combination of standard values from the literature and a deliberate design to maintain a relatively consistent average graph density across different topologies (except for fully connected / well-mixed population). This approach allows us to test performance across classic graph configurations while controlling for this key network property and ensuring the structures reflect plausible real-world characteristics (e.g., small-world, scale-free).
>
> In this work, we focus on structured populations and investigate how reputation and the corresponding information dissemination shape cooperative decision-making. This question is non-trivial: agents must jointly learn what "good" or "bad" means and how to act on different reputations based on local observation. To keep the spotlight firmly on this reputation-driven dynamics, we deliberately freeze the network. Static network structures allow us to disentangle the effect of norm emergence from the additional cooperation boost (or suppression) introduced by temporal networks, as recently highlighted in studies of dynamic graphs [1]. Once the reputation–cooperation interplay is understood in this cleaner setting, the natural next step is to let the graph itself evolve and examine how reputation and cooperation dynamics coevolve with the network topologies, a direction we explicitly reserve for future work.
>
> [1] Li, A., Zhou, L., Su, Q., Cornelius, S. P., Liu, Y. Y., Wang, L., & Levin, S. A. (2020). Evolution of cooperation on temporal networks. Nature communications, 11(1), 2259.
>
> To address the concerns about the population size, we have conducted new experiments with larger populations of n=30 and n=60 agents. The results of self-play experiment confirm that COOPER achieves a cooperation level that outperforms the baselines across all population sizes, but the cooperation ratio is lower for large populations, given the same sampling steps. To be more specific, with 30k steps, in scale-free networks, n=30 achieves an average cooperation ratio of 0.398, and n=60 achieves an average cooperation ratio of 0.336 (n=10 achieves an average cooperation ratio of 0.726). We speculate that the decline in cooperation is tied to gossip-efficiency differences across scales. In the 10-agent scale-free network generated by the Barabási–Albert model with m = 2, the average path length is only about 1.5-1.7 hops, so a reputation update reaches the whole population almost immediately. With 30 agents (same m = 2), the average path length grows to 2.4-2.6 hops (and for n=60, 2.9-3.1 steps), slowing convergence of the reputation estimates and weakening the signal that COOPER needs to sustain high cooperation levels.
>
> Additionally, we agree that a detailed investigation into how specific network characteristics (like graph density) influence reputation dynamics is a profound and important question. We have now acknowledged this explicitly in the revised manuscript as a key direction for future work.

---

> ### Author Response · Authors · 2025-11-26
>
> **Clarification of our baseline:**
>
> To ensure fair comparison, every baseline agent receives exactly the same raw observation as a COOPER agent. Hence, there is no difference in the MDP state/observation between methods. In baselines without reputation-related modules, such as PPO, reputation information is treated as just one more entry in the observation vector. Additionally, we want to clarify that our PPO baseline is not “vanilla” in the sense of lacking regularization. We keep the same actor–critic network width, entropy coefficient, clipping coefficient, learning rate, and so on, as COOPER; the only change is that the reputation assignment modules and the reputation-related loss are removed. We follow CleanRL’s PPO implementation and ensure that any performance gap comes solely from the reputation-related learning framework and module design.
>
> We believe these revisions tackle your core concerns. We will also revise the typos and polish our presentation according to the comments C5 and C6. Thank you again for helping us strengthen the paper.

---

### Official Review · Reviewer_56cV · 2025-10-25

**Soundness:** 1
**Presentation:** 4
**Contribution:** 2
**Rating:** 4
**Confidence:** 4

**Summary:**

This paper introduces COOPER, a reputation based reinforcement learning algorithm which is capable of finding cooperative solutions to social dilemmas by learning reputation assesment rules that are then used to condition agent's policies.  The method is composed of three primary modules: 1. a gossip-based reputation assesment module that aggregates the reputations of other agents (considered within a social network), 2. an interaction-based reputation assesment module that uses previous interactions to generate a reputation score, and 3. a policy network that conditions on these two assesments plus the history of interactions. Compared to previous works, the constituent modules of COOPER allow for the data to flow amongst them and overcome deep correlations between the policy and reputation modules, that often lead to instability issues like failure to converge. COOPER is shown to be able to learn sensible reputation norms in iterated donation games and the more challenging Coin Game.

**Strengths:**

I really appreciate the authors' efforts on the presentation of the manuscript. This work is extremely well written, and simple to understand. The method is also novel, compared to existing algorithms in the social dilemmas literature. Having established that these are the main strengths of the paper in more detail:

1. **Clarity**: The method is simple and presented such that each component is fully explained.  A good amount of the paper is spent on establishing the foundations necessary to understand the method, including other reputation-based methods.

2. **Relevance**: Social dilemmas are an often neglected subset of games that ironically show up in all sorts of real multi-agent interactions. I appreciate any efforts spent on designing algorithms that are capable of learning strategies with good performance on social dilemmas, especially those that do not rely on heuristics. In that regard, COOPER is a method that relaxes many of the heuristics of previous works.

3. **Novelty**: To the best of my knowledge there are no methods with a similar modular structure (for reputation-conditioned policies) as COOPER in the multi-agent reinforcement learning literature and more specifically, in the reputation-based literature.

**Weaknesses:**

The main concerns that I have regarding this paper is that the evaluation of of the algorithm leaves major holes and undermines the real usefulness of COOPER. I understand that previous works may have similar evaluation structures, but in my opinion showing that an algorithm is able to find cooperation by itself does not constitute a strong enough result. This is trivially achieved by summing the rewards of the players and running standard reinforcement learning algorithms. I am more interested in the reputation rules that COOPER learns, so here is my criticism: the learned reputation-based cooperation pattern (shown in Figures 7 and 8) do not correspond to well known solutions to social dilemmas, like tit-for-tat, which are well known Nash equilibria points of the game. In particular, these figures show that for fully connected social networks the agents converge to anti-tit-for-tat, a solution that is not a Nash and does not maximize social welfare. When the social network has limited connections the policy learned is close to always cooperate(which is exploitable), with the major caveat that the agent is less likely to cooperate if previously both of them cooperated. In other words, it finds exploitable solutions that are also Pareto sub-optimal.

With such underwhelming results it is hard for me to recommend acceptance unless the authors can address my concern, or explain if there is a flaw in my reasoning/interpretation.

On a separate note, there is an extensive line of Opponent Shaping literature that attempts to tackle social dilemma games [1, 2], including the very same Coin Game used as the hard evaluation benchmark of this paper. It is surprising to me that none of this literature is cited and that COOPER is not compared against, at least, some of these methods.

References:

[1] Foerster, J. N., Chen, R. Y., Al-Shedivat, M., Whiteson, S., Abbeel, P., & Mordatch, I. (2018). Learning with Opponent-Learning Awareness. In AAMAS 2018, 122–130.

[2] Duque, J. A., Aghajohari, M., Cooijmans, T., Ciuca, R., Zhang, T., Gidel, G., & Courville, A. (2024). Advantage Alignment Algorithms. arXiv:2406.14662.

**Questions:**

I am interested in comparing the performance of reputation-assessment methods in social dilemmas when compared to alternative approaches, so having said that,

1. Why did the authors not compare (baseline) with other works in the Opponent Shaping literature, which have shown to be capable of learning robust strategies in many social dilemmas including the Coin Game?
2. What is the authors' intuition on why the different social network topologies lead to substantially different reputation norms? How does for instance a sparsely connected network lead to a reputation norm that is more cooperative than that of a fully connected network (this result seems counterintuitive to me)?
3. How do the authors justify the convergence of COOPER to strategies that do not constitute Nash equilibria of the game?

I am willing to reconsider my score if some/most of these concerns are appropiately addressed.

---

> ### Author Response · Authors · 2025-11-26
>
> Thank you very much for your encouraging and thoughtful feedback. We are delighted to read that you found the manuscript clear, relevant, and novel.
>
> **Regarding the Comparison with Opponent Shaping Methods:**
>
> We agree that Opponent Shaping (OS) methods are highly relevant to social dilemmas and have shown promise in learning robust strategies. The primary reason for not including OS methods in our initial comparison was to focus on reputation-based mechanisms specifically. Our work aims to demonstrate the unique value of emergent reputation norms in promoting cooperation, distinct from other cooperation-enhancing mechanisms like opponent shaping.
>
> However, we acknowledge the importance of a comprehensive comparison. In response to your feedback, we have added a new experiment in self-play donation game comparing COOPER with two prominent OS methods: Learning with Opponent-Learning Awareness (LOLA) and Advantage Alignment Algorithms (AAA). The results will be included in Appendix G.
>
> **Regarding the explanation of higher cooperation ratio in scale-free networks:**
>
> In scale-free networks, hub agents play a crucial role. Hubs receive and propagate information from many sources, effectively acting as “reputation anchors”. Their high connectivity allows them to quickly establish and disseminate cooperative norms, influencing the behavior of peripheral agents. This leads to a more stable and cooperative environment overall. In contrast, fully connected networks lack such hierarchical influence. Every agent has equal influence, leading to noisy and inconsistent reputation updates. Without clear anchors, cooperative norms are harder to establish and maintain, resulting in more exploitable strategies.
>
> [1] Santos, F. C., & Pacheco, J. M. (2005). Scale-free networks provide a unifying framework for the emergence of cooperation. Physical review letters, 95(9), 098104.
>
> [2] Huang, L., & Han, W. (2025). Impact of heterogeneous network structures on the evolution of group behavior. Chaos, Solitons & Fractals, 200, 116946.
>
> **Additional Analysis of the Emerged Norm:**
>
> What COOPER learns is not anti-TFT. In the fully-connected setting the agents adopt an alternating C-D-C-D action sequence that is independent of the opponent’s last action (see figure 7 and Appendix F). Hence, it is a deterministic alternation that yields a mutual-cooperation payoff every second step, which is sufficient to keep the average return positive in the donation game (b=0.5, c=0.3).
>
> Under the learned reputation norm, this is a Nash Equilibrium: any unilateral deviation breaks the alternation and reduces the deviator’s cumulative payoff. We further analyzed the policy under different reputation values. The agent (as shown in figure 7) cooperates deterministically when the opponent’s reputation exceeds –0.3, and defects when it falls below –0.6. Between these thresholds, the probability of cooperation increases monotonically with the opponent’s reputation. Importantly, the agent’s own reputation also influences its behavior: higher self-reputation increases the tendency to cooperate at any given opponent reputation level. Action-based reputation updates are consistent: cooperation reduces the opponent’s reputation by $\approx$ 1, while defection increases it by $\approx$ 1.
>
> As for interaction-based reputation updates, cooperation reduces the opponent's reputation by $\approx$ 1, while defection increases it by $\approx 1$. We agree that these are not canonical game-theoretic equilibria, but they are emergent, sustainable, and computationally stable under decentralized learning with reputation. We will add these explanations to the manuscript (Section 5.3 and Appendix F) to better contextualize the learned norms.
>
> Let's consider a scenario with two players A and B.
>
> In the well-mixed population, the policy learned by COOPER is as follows:
>
> (1)$\xi>-0.3$ play C, $\xi<-0.6$ play D, $-0.6 <=\xi<= -0.3$ play C with $p(\xi)=\frac{\xi+0.6}{0.3}$
>
> (2)if A plays C, then $\xi_B$ -1, if A plays D, then $\xi_B$ + 1 (and same for B)
>
> Initially, let $\xi_A$=$\xi_B$=0. At step 1, A plays C, B plays C because $\xi_A=\xi_B > -0.3$. So, $r_A=r_B=0.2$. Then, update $\xi_A =\xi_B=-1$. At step 2, A and B both play D and then update $\xi_A=\xi_B=0$. And $r_A=r_B=0$. It is clear that the system is recursively running steps 1-2.
>
> But, if at step 1, let agent B deviate and play D, then $r_A=-0.3$, $r_B=0.5$. In this case, update $\xi_A=1$, $\xi_B=-1$. So, in the new step 2, A will play D and B will play C, so $r_A= 0.5$, $r_B=-0.3$. In this case, update $\xi_A=0$, $\xi_B=0$, and it goes back to the situation in step 1.
>
> It is clear that B has no gain in this deviation, so the policy learned by COOPER is stable.
>
> Once again, thank you for pinpointing both the merits and the shortcomings of our work. We are truly grateful for the time and expertise you devoted to strengthening our manuscript.

---

### Official Review · Reviewer_YN6o · 2025-10-31

**Soundness:** 1
**Presentation:** 1
**Contribution:** 1
**Rating:** 2
**Confidence:** 5

**Summary:**

The authors propose a distributed multi-agent reinforcement learning framework that jointly learns both reputation assessment rules and reputation-based policies directly from environmental rewards. The evaluation is based on the coin and donation game. The model used for the reputation-based mechanism is not completely clear - see comments below.

**Strengths:**

+ The problem of learning through emergent cooperation is an interesting one.

**Weaknesses:**

- The design of the loss function does not appear grounded on clear foundations, especially in terms of summative loss.
- How do you select the values of the weights? $\lambda_{env}$ and $\lambda_{conf}$?
- The mechanism of gossiping is not described in sufficient detail. Is it just one hop? Is it synchronous? It is also very unclear how the system is bootstrapped. How do you initialise the actual trust estimation? What is the impact of this initialisation on the results?
- The authors should rethink the notation that appears rather convoluted. For example, it is not necessary to write that a function $\psi_{\theta}$ is a function of a function of the node itself and its neighborhood, since the neighbourhood can be identified univocally from the node itself.
- The authors say that the loss in Equation (4) captures the human tendency to consider peers’ point of view, but it is very unclear why this might help for “emergent” reputation in the first place.
- The authors do not consider the impact of network size on the performance of their solution. In fact, it seems to me that that is an important dimension, especially considering different networks structures with the presence of hubs, such as Watts-Strogatz ones.

- The update presented in Section 5.2 is very difficult to understand and not clearly justified.

- In Section 5.2.2, the authors say it stimulates cooperation, but I think it is difficult to make that claim without comparing the results presented in the paper against a situation in which the mechanism is not present.

- The selection of the comparators is very difficult to justify. For example, the authors compare with a vanilla PPO, which appears not suitable/meaningful (it is designed for something completely different).

- The authors say that the negative reward is amplified. It seems to me that the positive is amplified as well given the design of the system. This part is not sufficiently justified.

- The discussion about the “leading eights” is unrelated to the rest of the paper (see Section 5.3.1).

- The ablation study presented in Section 5.3.2 is very difficult to understand. In fact, if you remove the evaluation mechanism, there is no information to be shared in the first place, I would say.

- Overall, given the information in the paper, it is rather difficult to reproduce the proposed solution. The authors do not report sufficient information to reproduce the proposed solution in my opinion, for example, in terms of parameters used for the simulations.

**Questions:**

- The authors should provide more information about the gossiping mechanism and its implementation.
- The author should clearly describe how the “trust” of another agent is estimated before being sent around. This is quite unclear.
- How is the system initialised?
- Can you explain the design of the gossip-based update (see Section 5.2)? How do you select $\delta$?
- The concept of self-play is very unclear in this context. Can you please provide more details?
- Why do you say that the negative reward is amplified and not the positive one (Section 5.3)?

**Details Of Ethics Concerns:**

None.

---

> ### Author Response · Authors · 2025-11-26
>
> Thank you very much for your detailed and constructive feedback. We appreciate your thoughtful questions and suggestions, which have helped us improve the clarity and completeness of our work. Below, we address each of your concerns point by point.
>
> **1. Loss Function Design**
>
> It is important to note that our method is built on the Proximal Policy Optimization (PPO) algorithm, following the implementation by CleanRL (https://docs.cleanrl.dev/rl-algorithms/ppo/). Therefore, the core structure of $L_{env}$ and $L_{ent}$ follows the standard PPO, which also motivates using PPO as one of our baselines in the empirical study. In other words, we extend the loss function of PPO with a socially-motivated regularizer $L_{conf}$ to support the emergence of consistent and meaningful reputation norms in multi-agent systems. Intuitively, if a group of agents shares similar evaluation criteria, their gossip becomes more informative and consistent. Thus, this alignment helps stabilize norm emergence and improves the reliability of reputation-based decisions. We have added a more detailed explanation and related reference (e.g., Pan et al., 2024) in Section 4 to support this design choice.
>
> Reference: X. Pan, V. Hsiao, D.S. Nau, & M.J. Gelfand, Explaining the evolution of gossip, Proc. Natl. Acad. Sci. U.S.A. 121 (9) e2214160121, https://doi.org/10.1073/pnas.2214160121 (2024).
>
> Moreover, summing heterogeneous objectives as shown in Equation (1) is standard practice in RL, and it allows the practitioner to trade off task reward against auxiliary desiderata, in our case, consensus among neighbours.
>
> **2. Hyperparameter Selection & Reproduction**
>
> In this paper, we do not have $\lambda_{env}$, and in practice, we tune $\lambda_{conf}$ empirically while $\lambda_{ent}$ is set to 0.05 as stated in Appendix B. In self-play settings, where agents learn from scratch, conformity is less critical, so we set $\lambda_{conf}=0$. In adaptation settings, where agents interact with rule-based agents with existing reputation norms, we set $\lambda_{conf}=0.5$ to encourage alignment with the existing norms. For a given environment, the best coefficient value can be selected via grid search over simulations.
>
> As for the rule-based reputation update, we set the parameter $\delta=0.25$. This prevents a single interaction from swinging the reputation score to the extremes (+1/-1), while still ensuring that meaningful differences emerge within a handful of encounters instead of dozens. To ensure reproducibility, we have added a more detailed description in Appendix B and will release the source code upon acceptance.
>
> **3. Gossip, Initialization, and Rule-based Agents**
>
> Gossip is synchronous and one-hop. More specifically, at each timestep, every agent shares its full reputation vector with all its neighbors simultaneously. Initially, reputation is set to 0 for all agents, since the reputation value ranges from -1 to 1, this initialization indicates a neutral and default starting point. For the rule-based agents, after each interaction, they adjust their assessment of the co-player by +/- $\delta$ based on whether the co-player cooperated or defected. Rule-based gossip is incorporated by averaging the reputation updates of neighbors.
>
> The initialization difference has some impact on the final stable state, in terms of the final cooperation ratio/reward, but it does not affect the pattern that COOPER learns to cooperate with emergent reputation. To be more specific, for self-play experiments on scale-free networks with 10 agents, initializing the reputation as all 0, all 1, all -1, and uniformly random results in the cooperation ratio of 0.726, 0.38, 0.56, and 0.48, respectively.
>
> | Initial $\xi$ | 30k cooperation ratio (scale-free, 10 agents, self-play) |
> |-----------|-------------------------------------------------------------|
> | all 0     | 0.726                                                       |
> | all +1    | 0.380                                                       |
> | all –1    | 0.560                                                       |
> | U[–1,1]   | 0.480                                                       |
>
> Here, initializing $\xi$ as all ones leads to the worst cooperation level. We conjecture that with \xi initialized as +1, the only possible update direction is downward, and once every reputation has been dragged slightly below +1, the population loses the numerical contrast needed to separate “good” from “bad”, and cooperation collapses. The fact that all -1 are better than all +1 indicates that there is a trend to assign a higher reputation to cooperators and a lower reputation to defectors, so starting with all -1 does not hinder the reputation rise for the cooperators. Moreover, initialization with 0 leaves the full range open where both positive and negative updates are feasible, and this symmetry produces the highest cooperation level.

---

> ### Author Response · Authors · 2025-11-26
>
> **4. Network Size and Scalability**
>
> We agree that network size is a critical dimension. In this work, we focus on proof-of-concept and design validation with small networks (n=10). We are willing to add additional experiments in Appendix E. More specifically, we repeat the self-play donation-game experiment with population sizes n = 10, 30, 60, while keeping the other network parameters the same. Due to the population changes, we correspondingly update the episode length to 50, 150, 300 steps (so that every two agents will have approximately 5 encounters in each episode). The results of the self-play experiment confirm that COOPER achieves a cooperation level that outperforms the baselines across all population sizes, but the cooperation ratio is lower for large populations, given the same sampling steps. To be more specific, with 30k steps, in scale-free networks, n=30 achieves an average cooperation ratio of 0.398, and n=60 achieves an average cooperation ratio of 0.336 (n=10 achieves an average cooperation ratio of 0.726).
>
> We speculate that the decline in cooperation is tied to gossip-efficiency differences across scales. In the 10-agent scale-free network generated by the Barabási–Albert model with m = 2, the average path length is only about 1.5-1.7 hops, so a reputation update reaches the whole population almost immediately. With 30 agents (same m = 2), the average path length grows to 2.4-2.6 hops (and for n=60, 2.9-3.1 steps), slowing convergence of the reputation estimates and weakening the signal that COOPER needs to sustain high cooperation levels.
>
> **5. Regarding the "amplified negative reward" Mentioned in Section 5.3**
>
> Under LR2, both positive and negative rewards are scaled by reputation. However, unilateral cooperation (which yields negative reward) becomes more penalizing for high-reputation agents, creating a disincentive to cooperate in uncertain environments. We have revised the expression to clarify this point and avoid suggesting that only negative rewards are amplified.
>
> **6. Self-play Definition**
>
> We apologize for the ambiguity. Following the standard MARL convention, self-play refers to a setting where all agents are COOPER agents, and they jointly learn from scratch without any pre-defined reputation rules or external supervision.
>
> **7.Ablation Study**
>
> As stated in Section 5.3.2, the ablation study is designed to test the individual contribution of $\psi$ and $\phi$. When $\phi$ is removed, agents still update reputations via gossip, but these updates are no longer grounded in interaction experience, leading to unreliable or stagnant beliefs. When $\psi$ is removed, agents rely only on local interaction history, which slows down norm propagation and limits social learning.
>
> **8.COOPER stimulates cooperation (Section 5.2.2)**
>
> In Section 5.2.2, learning agents (COOPER and the baselines) are adapting to a group of ALLD-RA agents with a discrimination bar of 0.5. Since ALLD-RA agents only cooperate if the opponent’s reputation is above 0.5, when initializing all agents’ reputation as 0, this group will converge to global defection. However, as shown in figure 4(b) and (c), COOPER’s cooperation leads to a positive payoff, indicating that ALLD-RA agents switch to cooperation with COOPER.The only way this can occur is that COOPER’s own reputation rises above 0.5 through its behaviour, thereby pulling the ALLD-RA agents above their threshold and triggering cooperative responses. This fact leads to the conclusion: COOPER stimulates cooperation.
>
> **9.Regarding the reason to mention "leading eight" norm**
>
> Finally, as mentioned in section 5.3.1, the "leading eight" norm is a widely discussed benchmark in evolutionary game theory for reputation-based cooperation[1,2]. We compare COOPER’s emergent norms to these rules to highlight the flexibility and structure-awareness of our learned norms.
>
> [1] Y. Murase, & C. Hilbe, Indirect reciprocity under opinion synchronization, Proc. Natl. Acad. Sci. U.S.A. 121 (48) e2418364121, https://doi.org/10.1073/pnas.2418364121 (2024).
>
> [2] Fujimoto, Y., & Ohtsuki, H. (2024). Who is a leader in the leading eight? Indirect reciprocity under private assessment. PRX Life, 2(2), 023009.
>
> **10.Regarding the notation**
>
> We thank the reviewer for pointing out the risk of notational clutter. We kept the explicit arguments in $u_i = \phi_{\theta_i}(\psi_{\theta_i}(\xi_i^t,\xi_{N_i}^t),H_i^t)$ because we wanted the reader to see, at a glance, which pieces of information are fed into each submodule. Simplifying this would make it harder to verify that the forward pass matches the architectural diagram in Figure 1. We are happy to streamline the notation in the final version, but we would prefer to retain this one-time explicit unpacking so that the data flow remains transparent.
>
> Once again, we sincerely appreciate your valuable feedback. We hope that the revised version addresses all your concerns.

---

### Official Review · Reviewer_g5o5 · 2025-11-03

**Soundness:** 3
**Presentation:** 2
**Contribution:** 3
**Rating:** 6
**Confidence:** 3

**Summary:**

This paper presents COOPER (COOPeration with Emergent Reputation), a novel distributed Multi-Agent Reinforcement Learning (MARL) algorithm designed to promote cooperation in social dilemmas by integrating emergent reputation mechanisms. The core contribution is the framework's ability to jointly learn a reputation assignment rule (reputation norm) and a reputation-based policy entirely from extrinsic environmental rewards. This approach addresses a key limitation of previous MARL methods that rely on predefined assessment rules (e.g., RR, IR) or use reputation as a fixed intrinsic reward signal (e.g., LR2), which often compromises adaptability and generalization. COOPER achieves this through a dual-module architecture: a gossip-based assessment component (ψ) aggregates neighbors' opinions, while an interaction-based assessment component (ϕ) refines beliefs based on direct experiences. A critical alternating optimization scheme manages the deep entanglement between the policy and reputation modules. Experiments conducted on the donation game and the coin game across various social networks (small-world, scale-free, fully connected) demonstrate COOPER’s efficacy in adapting to existing reputation norms and stimulating sustained cooperation in decentralized self-play scenarios. The results also offer insights into the co-emergence of cooperation and flexible, network-dependent reputation norms.

**Strengths:**

- Originality

The originality of this work stems primarily from successfully tackling the decentralized, joint learning of reputation assessment and reputation-based policy entirely from environmental feedback. Prior state-of-the-art methods typically assume a pre-existing reputation norm, which simplifies the learning process but results in a lack of adaptability when facing novel environments or unfamiliar agents. For instance, methods like RR rely on specific, predefined rules such as Stern Judging, while IR uses existing rule-based agents to foster norm learning. Furthermore, approaches like LR2 model reputation solely as an intrinsic reward signal, which can be sensitive to balancing extrinsic and intrinsic rewards and may lead to misaligned learning signals, particularly in challenging scenarios where unilateral cooperation is costly. COOPER differentiates itself by allowing the reputation norm to emerge from scratch within a fully decentralized setting, which is a significant conceptual advance in MARL.
The architectural design further contributes to originality by integrating two sophisticated components within the reputation assignment module: the gossip-based assessment (ψ) and the interaction-based assessment (ϕ). This dual structure is deliberately crafted to reflect how human societies balance aggregated social opinions with personal experiences, thereby enhancing robustness. The technical innovation to enable this co-learning is the alternating optimization scheme. By reversing the data flow during optimization, COOPER overcomes the latency and noise inherent in the feedback loop between reputation and policy, ensuring stable convergence guided purely by the extrinsic rewards.


- Quality

Technically, COOPER is founded upon the PPO algorithm but is substantially augmented with dedicated, differentiable reputation modules (ψ and ϕ). The framework is formalized through a partially observable Markov game, clearly defining the components necessary for reputation dynamics and network influence. The optimization scheme incorporates a theoretically justified conformity loss ($L_{conf}$), which acts as a graph-based smoothness prior to regularize ψ towards neighborhood consensus, improving sample efficiency without allowing gossip to entirely overpower direct experience, since ϕ retains the ability to refine beliefs based on fresh interaction data.
The empirical section is comprehensive, validating COOPER’s capabilities across critical axes. First, the adaptation experiments convincingly demonstrate COOPER’s robustness in non-self-play settings, showing that it can recognize and respond optimally to varied, pre-defined reputation policies like ALLD-RA and TFT-RA, often sustaining higher cooperation and reward levels than baselines. Second, the self-play results consistently establish COOPER's superior performance in establishing cooperation across small-world, scale-free, and fully connected network topologies, often succeeding where PPO agents converge to defection. Third, the ablation study provides crucial evidence that both the gossip (ψ) and interaction (ϕ) modules are necessary for robust cooperation; the removal of either component leads to a noticeable decline or complete failure of norm emergence. Finally, the detailed analysis of emergent norms lends credibility to the framework's ability to learn flexible, context-dependent social strategies.


- Clarity

The motivation for leveraging reputation is clearly linked to human social dynamics (gossip, long-term benefits) and the challenge of social dilemmas in autonomous systems. The mathematical formulation of the learning problem is clear, using the Partially Observable Markov Game (MG) framework.
Crucially, the paper successfully clarifies the complexity of the co-learning challenge and the devised solution. The distinct roles of the rollout phase (ψ→π→ϕ) and the optimization phase (ψ→ϕ→π) are explicitly noted, ensuring the reader understands how the method handles the feedback loop delay. The loss function components ($L_{env}$, $L_{conf}$, $L_{ent}$) are well-defined, with $L_{conf}$ being clearly positioned as a regularizer towards neighborhood consensus. Furthermore, the experimental section is highly accessible to the MARL community. The visualization of the emergent reputation norms using experience-to-action heatmaps (Figure 7, 8) effectively translates complex decentralized behavioral patterns into interpretable structures, explicitly showing how the learned norms differ based on network position (hub vs. leaf) and comparing them to theoretical models like the "leading eight" norms. The detailed appendices regarding implementation parameters (e.g., GAE lambda=0.95, PPO clipping=0.3) and network generation facilitate reproducibility.


- Significance

By enabling the decentralized, end-to-end emergence of reputation norms, COOPER addresses a foundational challenge in multi-agent cooperation, offering a mechanism that promotes farsighted behavior in mixed-motive games where myopic strategies typically lead to collective failure. This is evidenced by COOPER's ability to stabilize mutual cooperation, where baseline PPO agents fail.
The paper's findings establish a strong link between MARL agent behavior and established principles from evolutionary game theory. Specifically, the empirical observation that network heterogeneity (scale-free networks) supports the highest cooperation ratios, with "hub agents" acting as key cooperation anchors, directly aligns with theoretical expectations regarding network structure and cooperation promotion (e.g., Santos & Pacheco, 2005). The demonstration that a single COOPER agent can identify the reputation-based strategies of defecting populations (ALLD-RA) and stimulate a "cooperation cascade" that raises the average group reward is a powerful result, demonstrating real-world potential for social intervention in distributed systems. Finally, the approach provides a flexible framework that can adapt to diverse reputation norms and co-players, unlike rigid rule-based systems.

**Weaknesses:**

- Originality

While the core objective of joint learning is novel, certain aspects of COOPER rely on existing components or theoretical insights, tempering the claim of complete originality in all sub-modules. The underlying learning foundation is based on the Proximal Policy Optimization (PPO) algorithm, a standard technique, rather than a novel learning mechanism tailored exclusively for this reputation objective. The architectural separation into ψ (gossip/social assessment) and ϕ (direct interaction assessment) is conceptually motivated by long-standing theoretical models of reputation and indirect reciprocity in human societies.


- Quality

The fundamental reliance on truthful communication is a significant technical limitation: the model assumes that agents truthfully share their reputation assessments, and this assumption is neither relaxed nor tested in adversarial environments where agents might disseminate misinformation to manipulate reputations. Such a fragility limits the practical quality of the system in real-world deployments.

A second quality concern relates to the constrained memory capacity used in the empirical setup. Agents rely on a simple "memory-two" setting for interaction history, meaning reputation assignment is based solely on the last two interactions with a co-player. This simple constraint restricts the complexity of the reputation norms ϕ can learn and may fail to generalize when more sophisticated, long-term memory norms (e.g., those requiring knowledge of an agent's history over ten past interactions) are required to foster cooperation.


- Significance

The reputation metric is modeled as a simple scalar value $ξ_{i→j} ∈[−1,1]$. The authors identify this as a limitation, noting that it may lack the expressiveness required to capture complex behavioral nuances in more sophisticated environments, potentially resulting in oversimplified social evaluations. If reputation needs to be multidimensional (e.g., reputation for reliability, speed, fairness), the current scalar approach fundamentally limits the framework's direct applicability to many real-world coordination problems.

**Questions:**

1. The efficacy of COOPER assumes agents truthfully share their reputation assessments during the gossip phase (ψ). Given that reputation systems can be manipulated, how robust is COOPER to untruthful or adversarial gossip, and have the authors considered mechanisms to mitigate misinformation dissemination in future work?

2. In the ablation study, removing the interaction module (ϕ) entirely caused cooperation to "fail to emerge entirely," as updates remained unanchored. Does this imply that the conformity regularization alone is insufficient to provide a stable norm, even in highly connected networks (like the fully connected network)?

---

> ### Author Response · Authors · 2025-11-26
>
> Thank you very much for your insightful review and for raising important concerns regarding the robustness and practical applicability of COOPER. We appreciate your suggestions and have carefully considered them in our revision and planning for future work.
>
> **1. Regarding the robustness to untruthful or adversarial gossip:**
>
> You are right that assuming only truthful communication during the gossip phase ($\psi$) is a limitation in real-world deployments, as reputation systems are inherently vulnerable to manipulation. We fully agree and have explicitly identified this as a critical direction for future work in the Conclusion and Appendix H in the submission. However, the core contribution of COOPER is to show that a reputation norm and a reputation-based policy can co-evolve from scratch through the same reward signal. This issue is non-trivial even when communications are honest. Dropping the truthfulness constraint connects our methods to reality, but it still requires the same co-learning process.
>
> Besides, we have conducted preliminary experiments where agents are allowed to strategically manipulate the reputation information they share. Our initial findings suggest that such misinformation can indeed distort group-level cooperation dynamics, reducing group cooperation. In 10-agent self-play donation game experiments under scale-free networks, the cooperation level with strategic gossip is decreased from 0.726 to 0.47.
>
> To mitigate the impact of deceptive gossip, we can employ adversarial training within COOPER to help agents learn to detect and discount false reputation messages, or maintaining an explicit trust score for each neighbor that is updated based on how well their past gossip aligns with subsequent observations. These methods provide a straightforward starting point, though more sophisticated mechanisms may be investigated in future exploration. We will add the results and analysis to the final version in Appendix H.
>
> **2. Regarding the failure of cooperation when removing the interaction module, and the removal of conformity loss:**
>
> Yes, as shown in our ablation study, removing the interaction-based reputation update module ($\phi$) leads to a complete failure of cooperation emergence. This highlights a critical limitation of relying solely on the conformity loss $L_{conf}$ in Equation (1):
> $$L(\theta_i)=L_{env}(\theta_i)+\lambda_{conf}L_{conf}(\theta_i)+\lambda_{ent}L_{ent}(\theta_i)$$
>
> If we were to retain only $L_{conf}$, the learning objective would reduce to minimizing the deviation between an agent’s reputation assessments and the average of its neighbors'. While this encourages local consensus, it completely decouples reputation updates from actual interaction outcomes. Without environmental reward signals or direct interaction feedback, agents cannot validate or correct their assessments when the social consensus is misleading or outdated.
>
> **3.Regarding the memory constraint, less of expressiveness of scalar reputation, and the originality of COOPER:**
>
> While COOPER does build on familiar building blocks, i.e., PPO for policy optimisation and the long-standing idea that social gossip ($\psi$) and direct experience ($\phi$) jointly shape reputation, we believe its value lies in showing, for the first time, that these pieces can be incorporated into a fully-decentralized, end-to-end system that learns both what reputation means and how to act on it without any hard-coded assessment rule or extra reward channel. The challenge is that the policy needs believable reputations to act on, but the reputation module can only learn if the policy produces cooperative or defective outcomes whose consequences are delayed and noisy. By interleaving rollout order ($\psi \to \pi \to \phi$) with reversed gradient flow ($\psi \to \phi \to \pi$), we obtain a stable learning signal. The fact that the system learns a reputation norm in the well-mixed population, and a network variant norm in scale-free networks, suggests the architecture captures something fundamental about indirect reciprocity.
>
> On the issues of scalar reputations and “memory-two” design, we agree they are simplifications, but if cooperation can emerge under such conditions,  future works can follow up to determine how the memory design and reputation representation contribute to the performance. Actually, the modular design makes lifting these restrictions straightforward, i.e., $\phi$ can be parameterised as an LSTM or transformer to ingest arbitrarily long interaction traces, and $\xi$ can be replaced by a vector or graph embedding that carries separate dimensions for trustworthiness, fairness, etc. We are already exploring these extensions and hope the community will treat the present work as a reproducible starting point. Once again, we appreciate the careful reading and the push toward richer representations and longer memories. These ideas point exactly to the exciting next steps we hope this paper will motivate.

---

### Author Response · Authors · 2025-12-03
**Appreciation & Summary of our contribution and key revision**

**We sincerely thank all the reviewers, ACs, SACs, and PCs for investing your valuable time in reviewing our submission, especially during unexpected disruptions in this year’s reviewing process.**


All reviewers agreed that our work tackles an important and interesting open problem in multi-agent systems. **Reviewer g5o5** and **Reviewer 56cV** especially commended the manuscript’s clarity in presentation and the novelty of our proposed method. They wrote ***“the paper successfully clarifies the complexity of the co-learning challenge and the devised solution”***, ***“This work is extremely well written, and simple to understand. The method is also novel, compared to existing algorithms in the social dilemmas literature”***. **Reviewer YN6o** and **Reviewer 9icq** offered constructive guidance on strengthening the empirical design and promising future directions, e.g., extending to a dynamic network setting. We have incorporated every suggestion and performed additional experiments. The revised paper, we believe, resolves the raised concerns and is stronger than the original submission.

**Our Contribution:**

This paper introduces COOPER, a fully decentralized multi-agent reinforcement learning algorithm that, for the first time, jointly learns reputation assessment rules and reputation-based policies from scratch, using only environment rewards. Unlike prior approaches that rely on hand-crafted reputation norms or auxiliary reward signals, COOPER enables the co-evolution of reputation and cooperation through a novel modular architecture. It integrates gossip-based social influence ($\psi$) and interaction-based personal experience ($\phi$) in a carefully designed learning loop, allowing agents to overcome the challenges of partial observation and delayed feedback inherent in social dilemmas on a networked multi-agent system.

We have carried out extensive experiments to show that COOPER not only adapts to existing reputation norms when interacting with rule-based agents, but also spontaneously establishes cooperative norms in self-play, across diverse network topologies (small-world, scale-free, fully connected). The emerged norms are structure-aware, differing between hub and leaf agents in scale-free networks, and are more flexible compared to the theoretically grounded “leading eight” norms from evolutionary game theory. Importantly, COOPER’s modular and end-to-end design makes it extensible to richer reputation representations, longer memories, and dynamic network structures, positioning it as a general and principled foundation for studying and fostering cooperation in decentralized multi-agent systems.

**Key Revision:**

* In Appendix H, we have constructed additional experiments where agents can strategically gossip with neighbors (responding to **Reviewer g5o5**’s concern). COOPER still cultivates group cooperation, but the cooperation level is lower than that in a truthful communication situation. We have proposed adversarial training and neighbor-specific trust scores as possible ways to conquer this cooperation drop.
* To address **Reviewer YN6o**’s reproduction concerns, we have increased the number of experimental runs, documented the previously missing hyperparameters, and refined the initialization and gossip-update protocols in Appendices B & D, ensuring that every experiment can be replicated exactly. We will also release the source code upon acceptance.
* To resolve the **Reviewer YN6o** and **Reviewer 9icq**’s concerns regarding network size and topology parameters, we scaled self-play experiments to larger populations (n = 30, 60) and evaluated the performances under different reputation initialization to show that COOPER works throughout all these conditions; we also provide the principled rationale for every Barabási–Albert and Watts–Strogatz parameter in Appendix C and E.
* Responding to **Reviewer 56cV**, in Appendix G, we incorporated opponent-shaping baselines into the self-play comparison. Since these methods cannot properly deal with reputation information, COOPER outperforms them.
* In Appendix F, we supplemented a game-theoretic analysis showing that the norm COOPER converges to is a Nash equilibrium, guaranteeing stability against unilateral deviation. We believe this resolves **Reviewer 56cV**’s question regarding COOPER’s learned reputation norm and further strengthens our paper.
* We also polished wording throughout and added intuitive explanations and supporting references to avoid confusion.

---

### Note · Authors · 2026-01-16

I have read and agree with the venue's withdrawal policy on behalf of myself and my co-authors.